# Sensitivity of glacier volume change estimation to DEM void interpolation

Robert McNabb[1], Christopher Nuth[1], Andreas Kääb[1], and Luc Girod[1]

[1]Department of Geosciences, University of Oslo, Postboks 1047 Blindern, 0316 Oslo, Norway

**Correspondence:** Robert McNabb (robert.mcnabb@geo.uio.no)

**Abstract.** Glacier mass balance has been estimated on individual glacier and regional scales using repeat digital elevation models (DEMs). DEMs often have gaps in coverage ("voids"), the properties of which depend on the nature of the sensor used and the surface being measured. The way that these voids are accounted for has a direct impact on the estimate of geodetic glacier mass balance, though a systematic comparison of different proposed methods has been heretofore lacking. In this study, we determine the impact and sensitivity of void interpolation methods on estimates of volume change. Using two spatially complete, high-resolution DEMs over Southeast Alaska, USA, we artificially generate voids in one of the DEMs using correlation values derived from photogrammetric processing of Advanced Spaceborne Thermal Emission and Reflection Radiometer (ASTER) scenes. We then compare 11 different void interpolation methods on a glacier-by-glacier and regional basis. We find that a few methods introduce biases of up to 20% in the regional results, while other methods give results very close (<1% difference) to the true, non-voided volume change estimates. By comparing results from a few of the best-performing methods, an estimate of the uncertainty introduced by interpolating voids can be obtained. Finally, by increasing the number of voids, we show that with these best-performing methods, reliable estimates of glacier-wide volume change can be obtained, even with sparse DEM coverage.

## 1 Introduction

Glacier mass balance responds directly to climatic influences, and therefore long-term records of glacier mass balance reflect changes in climate. Traditional estimates of glacier mass balance have involved *in-situ* seasonal or annual measurement of accumulation and ablation at select locations, and extrapolation of these sparse measurements to the entire glacier (the glaciological method; see, e.g., Cogley, 2009). This can provide a temporally dense time series for an individual glacier, but for very large glaciers or at regional scales, it is neither practical nor even possible. As of January 2019, the World Glacier Monitoring Service has glaciological-method mass balance measurements for only 450 of the more then 200,000 glaciers worldwide (WGMS, 2019; RGI Consortium, 2017), the majority of which are performed on smaller glaciers, with mass balances that tend to be more negative than the regional average (e.g., Gardner et al., 2013).

More recently, glacier mass balance has been calculated over longer time spans and with larger spatial coverage by differencing remotely sensed surface elevation measurements of glaciers (e.g., Bamber and Rivera, 2007). Integrating these differences over the glacier produces an estimate of volume change. With careful consideration of the multi-annual surface change of snow,

firn, and ice composition (e.g., Huss, 2013), this so-called geodetic approach provides the total mass change of a glacier. The geodetic method can be used with both sparse measurements (as in the case of laser altimetry), or full-coverage measurements (as is often the case with DEMs). When the geodetic method is used with full-coverage datasets, it has been used to calibrate and/or validate time series of mass balance measurements that have been obtained through the glaciological method (Elsberg

et al., 2001; Zemp et al., 2010, 2013; Andreassen et al., 2016). With the current increase in the number of available, accurate Digital Elevation Models (DEMs) derived from airborne and in particular space-borne sensors, measurements of glacier mass balance using the geodetic method are and will be more prevalent, providing proper spatial accounting of the scale of glacier changes worldwide.

In this study, we focus on the estimation of geodetic mass balance from DEMs. In general, wide coverage DEMs are

created from sensors on aerial or satellite platforms falling into two categories, optical and radar. DEMs derived from optical sensors have the advantage of measuring the snow and ice surface directly, but data availability is subject to weather and light conditions, which can often be cloudy or dark in most glaciated regions around the globe. In addition, low-contrast areas on glacier surfaces, such as in the accumulation area, can often result in missing data or data voids, though this problem has been reduced with improved radiometric resolution of more modern sensors. DEMs derived from radar sensors are weather- and

illumination-independent, as the active sensor acquires data even through cloud cover and polar night. However, glaciers tend to occur in areas with steep and/or rough topography, and layover and shadow can confound efforts to unwrap elevations on glaciers (e.g., Rignot et al., 2001; Shugar et al., 2010). In addition, radar signals penetrate snow and ice differently, depending on the properties of the surface, as well as the frequency of the signal; this penetration results in a spatio-temporal systematic bias in surface measurement that is still poorly understood and constrained (e.g., Rignot et al., 2001; Dall et al., 2001; Gardelle

et al., 2012; Dehecq et al., 2016). DEMs derived from airborne laser scanning (e.g., Geist et al., 2005; Abermann et al., 2010; Andreassen et al., 2016) are highly accurate, spatially complete, and mostly avoid the penetration issues associated with radar-derived DEMs. Such DEMs are expensive to produce, however, and have similar requirements as optical sensors and aerial photography, i.e., clear sky or high clouds, conditions that can be difficult to find over glaciers.

In the most ideal scenarios to calculate geodetic mass balances from repeat DEM differencing, the entire glacier would be

sampled systematically and with similar accuracy. In the most commonly used DEMs for glacier mass balance described above, zones of missing data (hereafter called "voids") are rather common, and may severely bias estimates depending upon how these regions are accounted for (e.g., Kääb, 2008; Berthier et al., 2018). Several different methods have been applied in the literature, and we briefly summarize them here. They include bilinear interpolation of elevation or elevation differences (e.g., Kääb, 2008); filling with an average value from a surrounding neighborhood (e.g., Melkonian et al., 2013, 2014); multiplying the

average elevation change by the total glacier-covered area (e.g., Surazakov and Aizen, 2006; Paul and Haeberli, 2008; Fischer et al., 2015); and estimating elevation change as a function of elevation, integrating this curve with the glacier hypsometry (e.g., Arendt et al., 2002, 2006; Kohler et al., 2007; Berthier et al., 2010; Kronenberg et al., 2016). In addition, we can classify these methods into "global" (here meaning encompassing the whole of the dataset as opposed to worldwide) and "local" types, where "global" methods use data from an entire region or group of glaciers, while "local" methods fill voids using only information

from an individual glacier, or from data closely surrounding the voids.

While various methods are used in individual studies, the sensitivity of geodetic mass balance estimates to various interpolation methods is not clear. An overarching comparison of the numerous methods is lacking, and their subsequent effects on volume change estimates at both local and regional scales. Using correlation values derived from photogrammetric processing of optical stereo imagery, we artificially introduce voids into a high-quality, spatially-complete DEM, and difference this DEM to another spatially-complete DEM. We then apply 11 different methods to fill these artificially-produced voids, and compare the resulting estimates of volume change both glacier-by-glacier and regionally to determine the potential impact and sensitivity on volume change estimates. This study aims to quantify the effects of different void-handling approaches, and to suggest the void-handling methods best suited for accurate volume change estimation. As a final note, in this paper, we use two radar-derived DEMs: one derived from C-Band radar, and another derived from X- and P-Band radar. Biases in the derived volume change estimates exist due to differences in seasonal timing and radar penetration; as such, the estimates presented here should not be interpreted as mass balance estimates for these glaciers without additional corrections.

## 2  Data

### 2.1  Study Area

To test the impact of void interpolation methods on estimates of volume change, we chose the area surrounding Glacier Bay and Lynn Canal, Alaska, USA (Fig. 1). This area contains over 700 individual glaciers (Randolph Glacier Inventory (RGI) v6.0; Pfeffer et al., 2014; RGI Consortium, 2017), with glaciers ranging from sea level to over 4000 m a.s.l., covering an area of approximately 5900 km$^2$. Additionally, the region is home to a wide range of glacier types, including surge-type glaciers, retreating (and advancing) tidewater glaciers, and both large and small valley glaciers. As such, it is an ideal region to estimate the effects of using spatially incomplete DEMs to estimate glacier volume changes, as it provides a diverse sample of glacier types, sizes, and altitude ranges, with a high variability of intra- and inter-glacier elevation changes.

### 2.2  DEMs

#### 2.2.1  SRTM

We use the Shuttle Radar Topography Mission (SRTM) C-band global 1-arcsecond dataset as the reference DEM in this study. The SRTM was acquired in February 2000 aboard the Space Shuttle Endeavour, flying both C-band and X-band instruments (Van Zyl, 2001). This nearly global DEM is temporally consistent and therefore ideal and commonly used for geodetic mass balance estimation (e.g., Surazakov and Aizen, 2006; Larsen et al., 2007; Melkonian et al., 2013, 2014), though typically over longer time periods ($> 10$ year separation between DEMs). We have selected this dataset, and not the US National Elevation Dataset (NED) as other studies have used in the region (e.g., Arendt et al., 2002, 2006; Larsen et al., 2007; Berthier et al., 2010), as the NED DEM was produced by digitizing 1948 USGS contour maps (Larsen et al., 2007) which contained large biases at higher elevations on glaciers (see, e.g., Arendt et al., 2002, *supplemental material*).

Owing to the nature of the instrument, the acquisition, and the topography in the region, there are voids in the SRTM data on steep slopes due to shadowing and layover effects (e.g., Rignot et al., 2001). Filled SRTM products, such as the one distributed by CGIAR Consortium for Spatial Information (Jarvis et al., 2008), typically use the NED dataset to fill these voids, which can introduce significant anomalies and discontinuities into the on-glacier elevations. As these voids are typically small and confined to the glacier margins in steep-sloped areas, we used the non-void-filled SRTM dataset and update glacier areas in our calculations (when necessary) to ignore these no-data regions, essentially assuming they do not belong to the glacier; these voids correspond to $< 2.5\%$ of the on-glacier area for our glacier outlines and study region. These original SRTM voids will thus not affect our sensitivity analysis on estimates of volume change.

### 2.2.2 IfSAR

As part of the Statewide Digital Mapping Initiative, the State of Alaska is producing an interferometric synthetic aperture radar (IfSAR) DEM of the entire state. The data are acquired from airborne radar operating in X-band and P-band, and are provided in a native resolution of 5 m mosaics. In our study area, flights were flown in summer 2012 and 2013. These data are available from the U.S. Geological Survey (USGS, 2019). The DEMs have a reported accuracy of 3 m, though on low-angle slopes the provided metadata indicate an accuracy closer to ~1 m when compared to LiDAR swaths.

### 2.2.3 Glacier Outlines

We use the Randolph Glacier Inventory v6.0 data as a base to mask glaciers (RGI Consortium, 2017). The outlines in this region are mostly based on imagery from the mid-2000s, so we have manually updated the outlines using mostly cloud-free Landsat scenes acquired in late summer in 1999 and 2001, in order to ensure our glacier outlines correspond to the SRTM date as close as possible.

As the IfSAR DEMs are only available over Alaska, and not adjacent areas in British Columbia and Yukon, we have selected only glaciers that fall 90% or more by area within Alaska. Additionally, we have removed any glaciers where 10% or more of the glacier area is covered by both IfSAR collection years, in order to ensure that we are using temporally consistent data to estimate volume change. As voids over smaller glaciers result in more limited data to work with, and for our purposes it is better to have a large sample of on-glacier pixels, we also remove any glaciers with an area smaller than $1 \, \text{km}^2$. Finally, we remove any glaciers for which we did not have a result for all methods (i.e., where the glacier is completely voided). This results in a total of 415 individual glacier outlines used for the analysis.

## 3 Methods

We first calculate the "true" volume change by directly differencing the IfSAR and SRTM DEMs after co-registration following Nuth and Kääb (2011), and summing the elevation differences multiplied by pixel area within each glacier outline. We then calculate the regional volume change as the sum of these individual glacier volume changes, and estimate the uncertainty in the "true" volume change using the original (non-voided) IfSAR and SRTM DEMs. We introduce voids into the IfSAR

DEM, and interpolate those voids using each the interpolation methods described in Section 3.2. To assess the performance of each method, we compare the interpolated estimates to the "true" volume changes, and the uncertainties derived from the non-voided DEMs. The code used to generate and interpolate the DEM voids, as well as the results for each glacier, can be found at https://github.com/iamdonovan/dem_voids.

Ordinarily, using DEMs derived from radar of different bands, especially those acquired in different seasons such as the SRTM (February) and IfSAR (typically August/September), would require a consideration of the effects of differential radar penetration in snow and ice, as well as a temporal correction accounting for the difference in season, before converting elevation changes to a mass balance value (Haug et al., 2009; Kronenberg et al., 2016). In this region, the SRTM is known to have particularly high levels of penetration that cause significant biases when used in geodetic mass balance calculations (Berthier

et al., 2018). As our interest in this study is in isolating the effect of void interpolation methods on estimates of volume change, we ignore the differential penetration and temporal mismatch between our DEMs. We therefore highlight that biases will exist in the numbers provided in this study and stress that these estimates should not be interpreted as mass balances without significant additional corrections.

## 3.1   Artificial Void Generation

In order to investigate the effects of interpolating voids, we first simulate voids in the (arbitrarily chosen) IfSAR DEM to reflect the distribution and size of voids that might be expected in DEMs derived from optical stereo sensors. Correlation masks from 99 MicMac ASTER (MMASTER)-processed stereo scenes (Girod et al., 2017) provides the basis for void simulation as low correlation areas represent failure of the stereographic reconstruction and elevation determination. We thus use areas of low correlation in the ASTER scenes to mimic voids, providing a way to ensure that our artificial voids are similar to what would

normally be seen in DEMs derived from spaceborne optical stereo sensors.

    We average and mosaic the 99 ASTER correlation masks together, and select a correlation threshold of 50% to serve as the lower bound for acceptable data. This choice of threshold is based on a visual inspection of the mask produced, and the desire to mimic the ASTER data as much as possible. To further investigate the effects of interpolation method on the estimates of volume change, we also decrease the threshold to 35%, and increase it to 70%, 80%, 90%, and 95%, comparing the differences

for the best-performing interpolation methods. For each threshold value, we apply the resulting mask to the IfSAR DEMs, producing voids as shown in Fig. 2.

## 3.2   Void Interpolation

The following is a summary of the different methods used to fill the artificially-generated voids in the DEM and DEM difference (dDEM) products. We have split the methods into three general categories, "constant" interpolation, "spatial" interpolation, and

"hypsometric" interpolation.

### 3.2.1 Constant Methods

For the so-called "constant" interpolation methods, we calculate the mean (or median) elevation differences of the non-void pixels for each glacier, then multiply this value by the glacier area, thereby obtaining an average volume change for the glacier. Examples of this method in the literature include Surazakov and Aizen (2006); Paul and Haeberli (2008); Fischer et al. (2015).

### 5    3.2.2 Spatial Methods

1. *Interpolation of elevation*. This method interpolates raw elevation values of the surrounding pixels to fill voids. The resulting interpolated DEM is differenced from the second DEM, followed by calculation of the volume changes. Though Pieczonka and Bolch (2015) uses ordinary kriging to fill voids in the original DEMs, we choose to use bilinear interpolation for further comparison with the results of Kääb (2008). Examples of this approach can be found in Kääb (2008); 10    Pieczonka et al. (2013); Pieczonka and Bolch (2015).

2. *Interpolation of elevation differences*. Two original, unfilled DEMs are differenced to create a dDEM. Then, the voids in the dDEM are filled using bilinear interpolation. An example of this approach can be found in Kääb (2008); Zheng et al. (2018).

3. *Mean elevation difference in 1 km radius*. For each void pixel, we calculate the average elevation difference based on 15    on-glacier pixels within a 1 km radius of the void pixel. Examples of this approach can be found in Melkonian et al. (2013, 2014).

### 3.2.3 Hypsometric Methods

The so-called "hypsometric" methods are based on the assumption that there is a relationship between elevation change and elevation. They can be further sub-divided into "global" and "local" approaches, depending on whether the mean is calculated 20    using data from the entire dataset or area of interest (i.e., "global") or for an individual glacier only (i.e., "local"). We have chosen this terminology to be consistent with the terms used for other forms of interpolation. The global approach is usually used to extrapolate measurements from only a few glaciers to a regional scale (e.g., Arendt et al., 2002; Berthier et al., 2010; Kääb et al., 2012; Johnson et al., 2013; Nilsson et al., 2015), rather than to estimate an individual glacier's mass balance. Here, we use these methods to evaluate both individual and regional volume changes, in order to see the effects on individual glacier 25    changes.

1. *Mean (or median) elevation difference by elevation bin*. Here, the original, unfilled DEMs are differenced, and the entire dDEM is binned according to the elevation in the earliest DEM for each pixel within the glacier outlines. The mean (or median) elevation difference for each bin is then calculated and multiplied by the area of each elevation bin to get a volume change. The sum of the volume change of each individual bin then gives the volume change for the glacier. This 30    method is used by, e.g., Kääb (2008); Berthier et al. (2010); Gardelle et al. (2013); Papasodoro et al. (2015); Kronenberg et al. (2016); Brun et al. (2017); Dussaillant et al. (2018). If a glacier has an elevation range of 500 m or more, we use

m wide bins; otherwise we choose elevation bins that are 10% of the glacier elevation range. Additionally, where elevation bins are completely voided, we fill these bins using a third-order polynomial fit to the available data, so long as there is data over two thirds of the glacier elevation range.

2. *Polynomial fit to elevation difference by elevation bin.* The original, unfilled DEMs are differenced, and a polynomial is fit to the elevation differences as a function of the original elevation. This elevation curve is then integrated over the glacier hypsometery in order to calculate a volume change. Based on examples from the literature, such as Kääb (2008), we have chosen a third-order polynomial.

## 3.3 Uncertainties

To estimate the uncertainties in the true volume changes, we first co-register each DEM (SRTM, 2012 and 2013 IfSAR campaigns) to ICESat, using the method described by Nuth and Kääb (2011). We can then use the triangulation procedure described in Paul et al. (2017) to estimate the residual bias $\varepsilon_{\text{bias}}$ after co-registering the DEMs to each other; i.e. the uncertainty in correcting the mean bias between the DEMs. We also estimate the combined random error in elevation, $\varepsilon_{\text{rand}}$ by calculating the root mean square (RMS) difference of the population of dDEM pixels on stable ground. For each glacier, the error in volume change $\varepsilon_{\Delta V}$ can be estimated as:

$$\varepsilon_{\Delta V}^2 = (\varepsilon_{\Delta h} A)^2 + (\varepsilon_A \overline{\Delta h})^2, \tag{1}$$

with $A$ the glacier area, $\varepsilon_A$ the error in glacier area (here conservatively assumed to be 10%; Brun et al. (2017); Paul et al. (2017); Pfeffer et al. (2014)), and $\overline{\Delta h}$ the mean elevation change on the glacier. To account for spatial autocorrelation, as well as the two sources of uncertainty in the elevation differences ($\varepsilon_{\text{bias}}$ and $\varepsilon_{\text{rand}}$), $\varepsilon_{\Delta h}$ can be written:

$$\varepsilon_{\Delta h} = \sqrt{\frac{\varepsilon_{\text{rand}}^2}{\sqrt{n/(L/r)^2}} + \varepsilon_{\text{bias}}^2}, \tag{2}$$

where $n$ is the number of pixels (i.e., measurements) that fall within the glacier outline, $L$ is the autocorrelation distance (here assumed to be 500 m; Brun et al. (2017); Magnússon et al. (2016); Rolstad et al. (2009)), and $r$ is the pixel size (30 m). Finally, we can combine equations (1) and (2) to obtain:

$$\varepsilon_{\Delta V} = \sqrt{\left(A\sqrt{\frac{\varepsilon_{\text{rand}}^2}{\sqrt{n/(L/r)^2}} + \varepsilon_{\text{bias}}^2}\right)^2 + \left(\varepsilon_A \overline{\Delta h}\right)^2}. \tag{3}$$

## 4 Results and Discussion

### 4.1 Void Distribution

Fig. 3 shows the void and area frequency distributions per normalized glacier elevation bin. For the 50% threshold case, most glaciers (64.4%, 268 glaciers) have a total void percentage below 20%, with only a small number (6.7%, 28 glaciers) having

more than 40% voids. Voids are distributed similarly to glacier area with respect to normalized glacier elevation (i.e., the elevation divided by the elevation range), and most of the voids, as well as most of the glacier area, are found in the middle third of the glacier elevation range. This portion of the range corresponds to the flatter, mostly featureless portions of the accumulation area, which leads to lower correlation in the ASTER scenes. These void and area distributions, along with the range of elevation changes, suggests that the middle third of the glacier elevation range is the most important to ensure correct estimation; that is, uncertainties introduced by interpolating over voids in the upper and lower thirds of the elevation range will be muted, owing to the typically smaller areas and percentage of voids in these ranges.

## 4.2   Variability of elevation change

Fig. 3 also shows the mean and median elevation changes per normalized elevation bin, and the standard deviation of elevation changes. The highest elevation change variability on-glacier is in the lower portion of the glacier, where most of the dynamic change is occurring. Higher up in the accumulation area, the variability is much lower, and the mean differences are close to zero. These general patterns can be observed in the spatial distribution of elevation changes shown in Fig. 4. In general, elevation changes in the region are negative, especially at lower elevations, as noted in other studies (e.g., Larsen et al., 2007; Johnson et al., 2013; Melkonian et al., 2014; Berthier et al., 2018). Some exceptions include Margerie (RGI ID: RGI60-01.20891), Johns Hopkins (RGI60-01.20734), and Rendu Glaciers (RGI60-01.21013) in the 2012 acquisition area, and Taku Glacier (RGI60-01.01390) in the 2013 acquisition area (see Fig. 1 for glacier locations). Margerie, Johns Hopkins, and Taku Glaciers are some of the few currently advancing tidewater glaciers in Alaska (e.g., Motyka and Echelmeyer, 2003; Truffer et al., 2009; McNabb and Hock, 2014), while Rendu Glacier has been previously identified as a surge-type glacier (Field, 1969). The pattern of elevation change shown on Rendu Glacier in Fig. 4, with thinning at higher elevations and pronounced thickening at lower elevations, is suggestive of a surge sometime between February 2000 and August 2012 (e.g., Raymond, 1987; Björnsson et al., 2003); a tributary of Margerie Glacier also appears to have surged during this time period.

The variability of elevation changes in the region is quite high, with a standard deviation of on-glacier elevation changes of $0.65\,\mathrm{m\,a^{-1}}$ for all glaciers in the region. This level of variability significantly smaller than other parts of Alaska ($1.14\,\mathrm{m\,a^{-1}}$ for glaciers in Western Alaska; Le Bris and Paul, 2015), but significantly higher than regions in High Mountain Asia, where Brun et al. (2017) found intra-regional standard deviations in mass balance of $\sim$0.2 m.w.e $\mathrm{a^{-1}}$ ($\sim$0.24 m $\mathrm{a^{-1}}$ given their assumed density of $850\pm60\,\mathrm{kg\,m^3}$). Compared to values estimated from ICESat (Nilsson et al., 2015), this region is in line with Svalbard ($0.7\,\mathrm{m\,a^{-1}}$), higher than the Canadian Arctic ($0.34/0.42\,\mathrm{m\,a^{-1}}$ for North and South, respectively), and much lower than Iceland ($1.14\,\mathrm{m\,a^{-1}}$).

## 4.3   Impact of void interpolation on Individual Glacier Estimates

The variability of elevation gain and elevation loss shown in Fig. 4 informs some of the patterns shown in Fig. 5. Generally, the global hypsometric methods are the farthest from the true values, which is perhaps not surprising in a region with a high variability of glacier elevation changes. Glaciers that are far from the mean of the glacier-wide average elevation changes (-$0.36\,\mathrm{m\,a^{-1}}$) will tend to be far from the true volume change when the volume change is estimated using regional values, as the

data used for the interpolation do not reflect conditions at that particular glacier. Thus, interpolated estimates for glaciers losing much more than the regional average tends to be overestimated (i.e., less negative change), while interpolated estimates at glaciers that are losing less than the average, or even increasing in volume, tends to be underestimated (i.e., more negative/less positive change). Methods which use data from an individual glacier, or in a small area close to the particular glacier outline, tend to do a much better job of reproducing volume changes over each of these glaciers than do these global methods.

The statistical summary for the difference in volume change estimates over all glaciers individually (Table 1) shows that on average, mean and median differences to the true values are generally low ($<0.1\,\mathrm{m\,a^{-1}}$), as are RMS values (typically $<0.2\,\mathrm{m\,a^{-1}}$ with the exception of the global hypsometric methods). The percentage of estimates that fall within the uncertainty range of the true volume change estimates for most of the methods is quite high, above 95%. One notable exception is the constant median method described in section 3.2.1, which aside from the global hypsometric methods, shows the fewest number of glaciers for which the interpolated value falls within the uncertainty (84%), the largest individual overestimation at $1.22\,\mathrm{m\,a^{-1}}$, the largest mean and standard deviation ($0.07\pm0.19\,\mathrm{m\,a^{-1}}$), the largest RMS difference ($0.20\,\mathrm{m\,a^{-1}}$), and the worst agreement with the regional volume change estimate ($0.69\,\mathrm{km^3\,a^{-1}}$ overestimation).

Fig. 6 shows the elevation change over Taku Glacier, with voids filled in for the nine non-constant methods. The spatial interpolation methods (Fig. 6b-d) and the local hypsometric methods (Fig. 6h-j) show the most similarity to the original elevation changes (Fig. 6a), with some subtle differences. The hypsometric methods have the effect of smoothing out the patterns of elevation change, whereas the spatial interpolation methods tend to preserve the original spatial patterns within elevation bands. Near the dividing lines between glaciers, discontinuities can be seen in the local hypsometric maps, compared to the more gradual changes across dividing lines seen in the original elevation changes and the spatially-interpolated maps. This suggests that the choice of glacier outlines can have an impact on the resulting volume change estimates. Finally, the global hypsometric methods (Fig. 6e-g), taking data from the region, do not faithfully reproduce the anomalous elevation change patterns for Taku Glacier.

For the largest 20 glaciers in the dataset (all $>100\,\mathrm{km^2}$), which represent 61% of the total glacier area for the glaciers studied, as well as 68.8% of the volume change in the region (a total of -49.9$\,\mathrm{km^3}$), we see a number of patterns related to each of the methods. Fig. 7 shows that for these largest glaciers, most of the methods fall within $\sim\pm0.3\,\mathrm{m\,a^{-1}}$ of the true value, with significant outliers for some methods on some glaciers. For example, each of the methods for Hole-in-the-Wall Glacier (RGI60-01.27102) are clustered quite close to the true value, with the exception of the global methods. Hole-in-the-Wall Glacier is directly adjacent to Taku Glacier, and is also slightly gaining mass, thus leading to the discrepancy with the regional averages. In general, the global methods under- or over- estimate volume change, with only a few cases where the results within the uncertainty of the true value. For another glacier, Riggs Glacier (RGI60-01.21001), the non-global methods give a value within $\sim0.05\,\mathrm{m\,a^{-1}}$ of the true value, while the global methods are still within $\sim0.25\,\mathrm{m\,a^{-1}}$; the number of voids induced on this glacier are relatively small (19% of the glacier area), and the elevation changes on this glacier are also similar to the regional ones (strong elevation loss at lower elevations, small gain at higher elevations).

Fig. 8 shows a box plot of the distribution of differences to truth for each method, using both the largest glaciers (Fig. 8a), and all glaciers (Fig. 8b). Based on the size of the interquartile range and the mean difference of each interpolated estimate, the

best-performing methods are the spatial interpolation methods, the local hypsometric method, and the constant mean method, for both sets of glaciers. For all glaciers, the outlier range is also the smallest for the local hypsometric methods, linear interpolation of elevation change, constant mean, and the 1 km average approach.

Table 2 shows the differences to the true volume change for two glaciers with some of the largest deviations from the true values. The global methods for Taku Glacier have some of the largest negative changes, all below -0.50 m a$^{-1}$. These differences are most likely for the reasons discussed above: the data being used to interpolate voids for Taku Glacier are far more negative than reality. The constant median estimate for Field Glacier (RGI60-01.01520) has the largest overall change from the true value for the non-global methods, at 0.92 m a$^{-1}$. For this glacier, only the constant median method and the global methods perform particularly poorly; the rest are all within $\pm$0.15 m a$^{-1}$ of the true estimate of -1.15 m a$^{-1}$ ($\sim$13%). As shown in Fig. 9, this is most likely because of the heavy slant towards very negative elevation changes in the elevation change distribution for Field Glacier. While representing significantly more of the glacier area, the values near 0 m elevation change found on the glacier are small compared to the extremely negative values, and so the median is pulled heavily towards zero, which greatly underestimates the volume change.

## 4.4  Impact of void interpolation on Regional Total

While the differences to the true values, when averaged over all glaciers, tends to be close to zero, the differences in the regional estimates can vary substantially, as shown in Table 1. The methods that came closest to the "true" volume change for the region were local mean hypsometric method, linear interpolation of elevation differences, and the global mean hypsometric method, which all yielded estimates within 0.03 km$^3$ a$^{-1}$ (0.8%) of the regional total. The global mean hypsometric method is often used in altimetry-based studies to extrapolate measurements to unsurveyed glaciers, either using absolute or normalized elevation (e.g., Arendt et al., 2002; Kääb et al., 2012; Johnson et al., 2013; Larsen et al., 2015), and our results indicate that relatively little bias is introduced to the regional estimate through this form of extrapolation, at least for this example. In this study, we have used absolute elevations, rather than normalized elevations, for the global methods. In other regions, it may be worth comparing the differences between using absolute and normalized elevations.

The next best estimates after the three closest were the local median and polynomial hypsometric methods, linear interpolation of elevation, and the 1 km average method, all coming within 0.15 km$^3$ a$^{-1}$ (4%) of the regional total. One explanation for the overall worse performance of the elevation interpolation method versus linear interpolation of elevation change is discussed in Kääb (2008): elevations on the glacier surface are not necessarily self-similar in a given area, and elevations can vary greatly even on relatively small length scales. As for the 1 km average method, it may be that 1 km is too large of an area to try to average over for some glaciers in this region, or it may be that the average window used is including values from neighboring glaciers that have very different patterns of elevation change, thus behaving more like a "global" method in some areas.

The methods that came the farthest from the regional total were the constant median method and the global polynomial hypsometric method, both overestimating the regional total volume change by over 0.5 km$^3$ a$^{-1}$, well above the uncertainty of 0.4 km$^3$ a$^{-1}$. The constant median had the worst agreement with the regional total, at 0.69 km$^3$ a$^{-1}$ (18%). In contrast to this, estimating volume changes using the global median hypsometric method underestimated the regional total volume change by

$0.37\,\mathrm{km^3\,a^{-1}}$. While for an entire glacier, the median elevation change skews very heavily towards zero due to the asymmetry in positive and negative values of elevation change, this is not necessarily the case for an elevation bin. As noted by Kääb (2008), and borne out by the elevation change interpolation method, elevation changes tend to be rather self-similar on small spatial scales, and the median change for an elevation bin tends to be a more accurate reflection of the actual elevation change. That said, for both the local and global methods, using the mean rather than the median yields a better result on both an individual and regional basis, suggesting that the mean is more representative as a rule.

## 4.5   Increasing void area

To estimate the sensitivity of the different methods to the amount of voids in the DEMs, we varied the correlation threshold from 35% to 95%. The effect of using each threshold on the mean percent void for all glaciers is shown in Fig. 10. Above a threshold of 70%, the mean void percentage per glacier increases dramatically, up to 75% voids when using a threshold of 95%. In the following analysis, we compare the total set of interpolated volume changes for all glaciers over all threshold scenarios. We have limited the number of methods discussed to those that performed the best in the 50% threshold case (described in sections 4.3 and 4.4): the constant mean method, linear interpolation of elevation changes, the 1 km average method, the global mean hypsometric method, and the local mean and median hypsometric methods.

Fig. 11 shows that the local hypsometric and spatial interpolation methods can tolerate a high void percentage (>50%) before more than 10% of the estimates fall outside of the uncertainty range; beyond 70% voids, this percentage increases dramatically. The constant mean method does not perform as well for lower void percentages, but it does not drop as quickly at higher void percentages as the other methods. As expected, the global mean hypsometric method has low values throughout the range of void percentages, though there is not as much dependence on the void percentage as with the other methods. Even at the highest void percentages, the spatial interpolation methods perform remarkably well, with upwards of 75% of estimates falling below the uncertainty range for both linear interpolation of elevation change and the 1 km average; this may not be true for other datasets, which we discuss more in section 4.6.

Based on the results for the 50% threshold case, we compared the three methods which gave the best results on a regional basis: linear interpolation of elevation change and the global and local mean hypsometric methods, with results shown in Fig. 12. The global mean hypsometric method shows little variation overall, with differences to truth generally negative and with a large standard deviation. As shown in Fig. 11, linear interpolation of elevation change and the local mean hypsometric method remain close to the true values of volume change up to around 50-60% voids, before the standard deviation increases dramatically, though less so for linear interpolation of elevation change. As the local hypsometric method requires data in a given elevation bin for interpolation, it makes sense that with higher void percentage, the interpolated estimates are further and further from the true values of volume change, while linear interpolation requires less data to provide an estimate; as long as the missing values are similar enough to the non-voided values, the interpolated estimates of volume change do not deviate significantly from the true volume changes.

## 4.6 ASTER differences

While linear interpolation of elevation differences performs very well with these DEMs and voids, it should be noted that these DEMs are very smooth, without significant noise in elevation values over glacier surfaces. When using DEMs that are noisier, this method may actually perform worse, as it may amplify this noise. To illustrate this, we used ASTER DEMs acquired on 13 August 2015 over a portion of the 2012 IfSAR acquisition area, and differenced these DEMs to the SRTM. The ASTER DEMs were processed using MMASTER, and along-track and cross-track biases were corrected using the 2012 IfSAR DEM (see Girod et al., 2017, for more details on these corrections).

Compared to the IfSAR DEM, the ASTER DEMs are quite noisy in the accumulation areas of glaciers, owing to the low contrast, and hence low correlation, between the original images in the ASTER scenes. As such, even after correlation masking, there is significant noise in the DEM difference map (Fig. 13a). When these values are linearly interpolated, the resulting dDEM shows clear interpolation artifacts and elevation changes that differ greatly from the original IfSAR/SRTM differences (Fig. 13b), biasing the estimated volume changes.

For the 91 glaciers covered by these ASTER DEMs, the other best estimates named above (local mean hypsometric and global mean hypsometric) yield a total volume change estimate of $\sim 0\,\mathrm{m\,a^{-1}}$, whereas linear interpolation of elevation differences yields a volume change of $+0.2\,\mathrm{m\,a^{-1}}$. Looking further at this, this discrepancy is almost entirely due to one glacier, Johns Hopkins Glacier. Linear interpolation of elevation changes yields a volume change estimate of $\sim 1\,\mathrm{m\,a^{-1}}$ for this glacier, while the other estimates yield values of $\sim 0.6\,\mathrm{m\,a^{-1}}$.

Thus, we caution against using a direct linear interpolation of elevation differences to fill voids without first filtering or otherwise removing potential outliers, which are often located near voids, which increases their influence in a linear interpolation. We also caution against using this approach when the distances between known values are quite large in relation to the glacier width. The local mean hypsometric approach used by many studies performs just as well as linear interpolation of elevation differences in the idealized case analyzed here, and therefore appears to be more robust against this kind of noise, and is easily implemented in place of linear interpolation.

A question then arises: when using noisy, 'real-world' data such as ASTER DEMs, is it better to keep only the most reliable values for a given DEM, potentially producing large voids that must be interpolated, or is it better to have a more complete DEM? Given the results presented in this section, and the results shown in section 4.5, we suggest that on-glacier areas with relatively low correlation (i.e., reliability) can still have usable data. As the local hypsometric methods tend to be more robust against noise, and can tolerate a rather high percentage of data voids (up to $\sim 60\%$; Fig. 12), a strategy of using lower correlation thresholds ($\sim$50-60%) in combination with the local hypsometric approach seems well-suited to making the most use of the available data.

## 5 Conclusions

We have compared 11 different methods for interpolating voids in DEM difference maps over glaciers, and compared the effects of these different methods on estimates of glacier volume change. Two methods, linearly interpolating elevation changes and

the local mean hypsometric method, performed well on an individual glacier basis, producing estimates within the uncertainty of the original estimates. These two methods, as well as a third, the global mean hypsometric method, also performed quite well in estimating the regional total volume change, differing from the true estimate by less than one percent. For the low-noise level DEMs we have used, linearly interpolating elevation differences tends to produce elevation change maps that look the most

similar to the original maps. This may not hold, however, for voids that take up a larger portion of the glacier area, where the assumption that elevation changes are similar over small distances may be violated, and interpolation artifacts would introduce larger uncertainties. Additionally, this may not hold for DEMs that are noisier, especially in low-contrast areas such as the accumulation zones of glaciers; as such, we caution against adopting this method without first considering the characteristics of the DEMs being used. In terms of individual glacier estimates, the local mean hypsometric method performs quite well and

appears more robust in the face of noisy DEMs, which perhaps explains its widespread use in studies of glacier volume change and geodetic mass balance.

On average, most of the methods perform well, with low mean, median, and RMS differences for all methods, though large outliers skew the differences in the regional totals. The constant median method, however, tends to work quite poorly, owing to the asymmetrical distribution of positive and negative elevation change values; i.e., the glaciers in the region tend to have

significantly more negative values of elevation change than positive values. Unless there is good reason to think the distribution of elevation changes for a particular glacier or region is more symmetrical, this method should be avoided. The same can be said for using a median hypsometric approach, which does not perform as well as the mean hypsometric approaches. As might be expected, using regional data to estimate the volume change of an individual glacier quite often performs poorly, though the regional total volume change can be well-approximated in this way.

The bias introduced by a given method is also dependent on the size of the data voids. For the two most accurate methods on both an individual glacier and regional basis, interpolating voids of up to 50% tends to introduce small differences in the estimated volume change; most of the change is happening in the lower parts of the glacier where the void percentage is smaller. Above 60-70% voids, however, the errors grow substantially, and are usually significantly higher than the uncertainty in the original datasets. This is not the case for the global interpolation methods, however, which have large errors for individual

glaciers that are mostly independent of the void percentage. Thus, the void percentage does not have as pronounced an effect on the regional total estimated using the global mean hypsometric method, implying that its use in regional-scale altimetry studies is well-founded.

In summary, the effect of DEM voids on estimates of geodetic mass balance depends on the size of the voids, the magnitude and spatial pattern of changes on the glaciers, as well as the nature of the DEMs used. The choice of void interpolation method

is important, and if not considered properly, biases many times the uncertainty of the volume change measurement can be induced. On the regional scale, biases of up to 20% can be induced. The choice of "best method" will depend on the ultimate goal of the study, as well as the nature of the voids in the DEMs and the changes of the glaciers. Interpolation methods using elevation differences from an individual glacier, or differences within a close proximity to an individual glacier (in the case of glacier complexes), tend to be the most accurate and robust. If the DEMs used have significant noise, or have large voids,

however, linear interpolation may not be suitable. If attempting to estimate geodetic mass balance for unsurveyed glaciers,

as is needed in many altimetry-based studies, only a global method will suffice, though the mass balance estimate for a given unsurveyed glacier should not be taken at face value. Additionally, the regional estimate for such a case may be strongly biased. As each of these different methods are relatively easily implemented, however, a comparison of the different methods should be attempted in order to provide a measure of the uncertainty introduced by interpolating voids in the data.

5  *Code availability.* The code used to generate and fill voids, as well as the resulting data for each glacier, can be found in a git repository at https://github.com/iamdonovan/dem_voids

*Competing interests.* The authors declare that no competing interests are present.

*Acknowledgements.* This study was funded in part by the ESA project Glaciers_cci (4000109873/14/I-NB) and the European Research Council under the European Union's Seventh Framework Programme (FP/2007–2013)/ERC grant agreement no. 320816. Alaska IfSAR

10  DEM data are provided through State of Alaska's Statewide Digital Mapping Initiative and the US Geological Survey's Alaska Mapping Initiative. Both the IfSAR and the SRTM DEM are distributed through the USGS Earth Resources Observation Center. Acquisition of ASTER images was guided by NASA JPL, though the ASTER science team and the Global Land Ice Measurements from Space (GLIMS) initiative. The authors wish to thank Frank Paul and Stephen Plummer for their helpful comments on early versions of the manuscript. Nick Barrand and two anonymous reviewers provided many insightful comments which helped improve the quality of the manuscript.

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

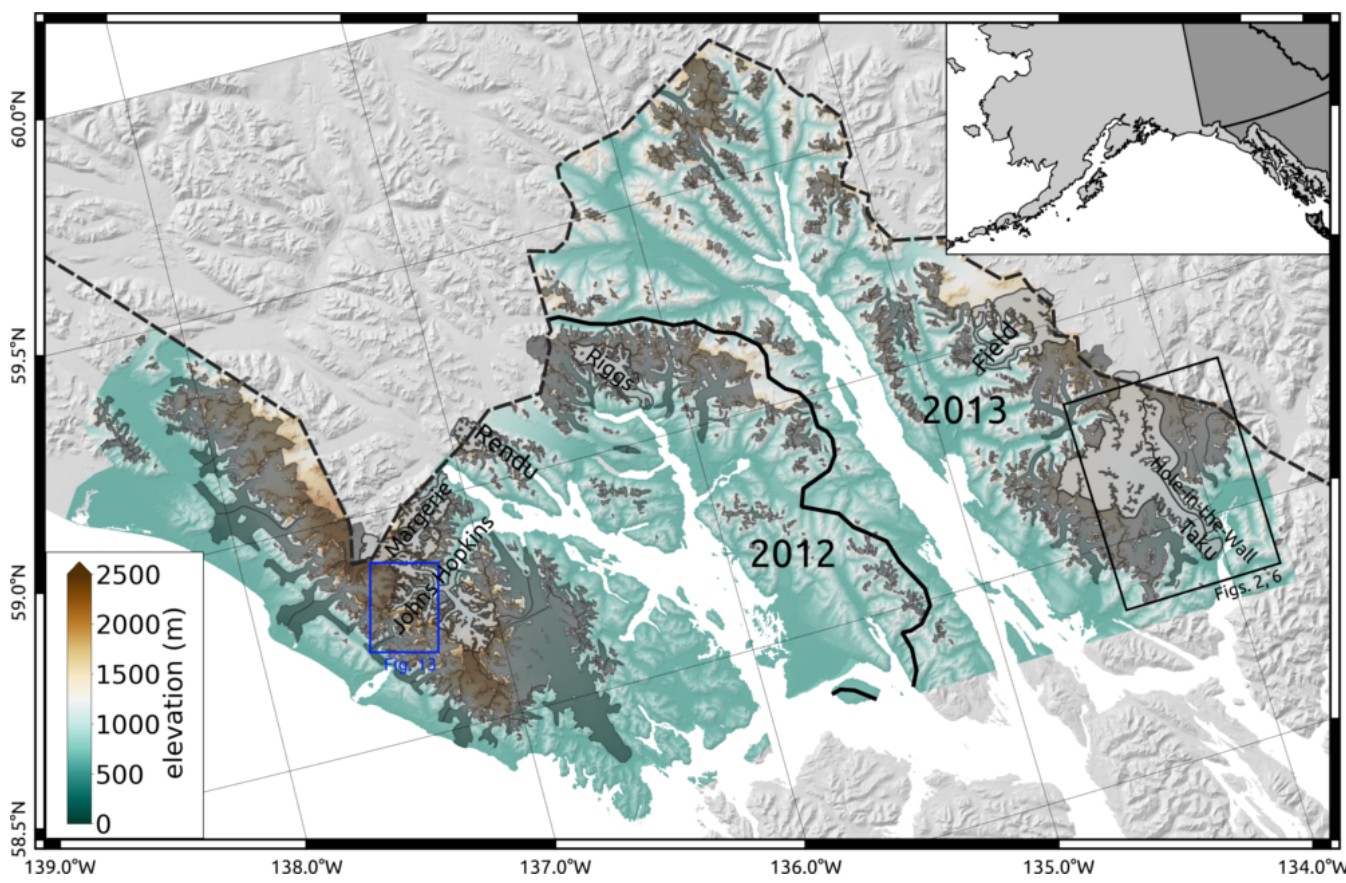

**Figure 1.** Study area in Southeast Alaska, USA. IfSAR DEMs are displayed overtop SRTM hillshade. Solid black line indicates boundary between 2012 and 2013 IfSAR acquisitions. Dashed black line indicates US-Canada border. Dark gray outlines show glacier extents; named glaciers (light gray outlines) are discussed further in the following sections. Black outline indicates extents of Figs. 2 and 6; blue outline indicates extent of Fig. 13.

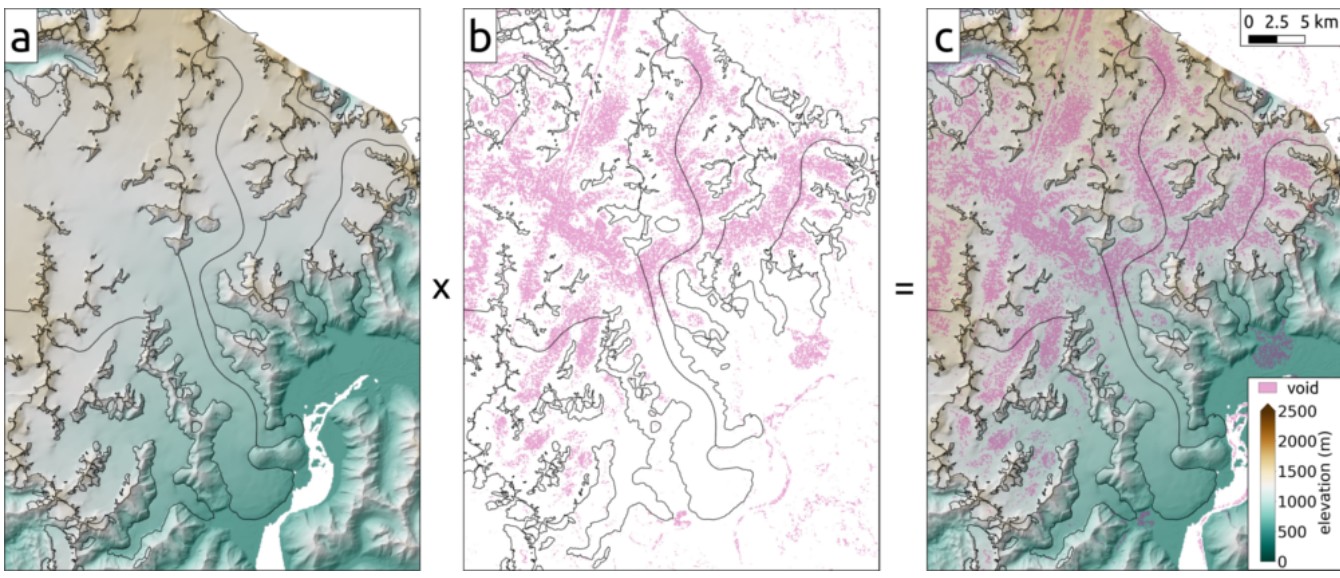

**Figure 2.** Example of masking procedure over Taku Glacier, with RGI outlines shown in black. The IfSAR DEM (a) is masked using the composite correlation mask from the ASTER products with a correlation threshold (here 50%) (b), to produce a DEM (c) with voids (in purple) similar to expected voids in an optical DEM. In the middle panel, purple represents areas where the ASTER correlation is below the chosen threshold.

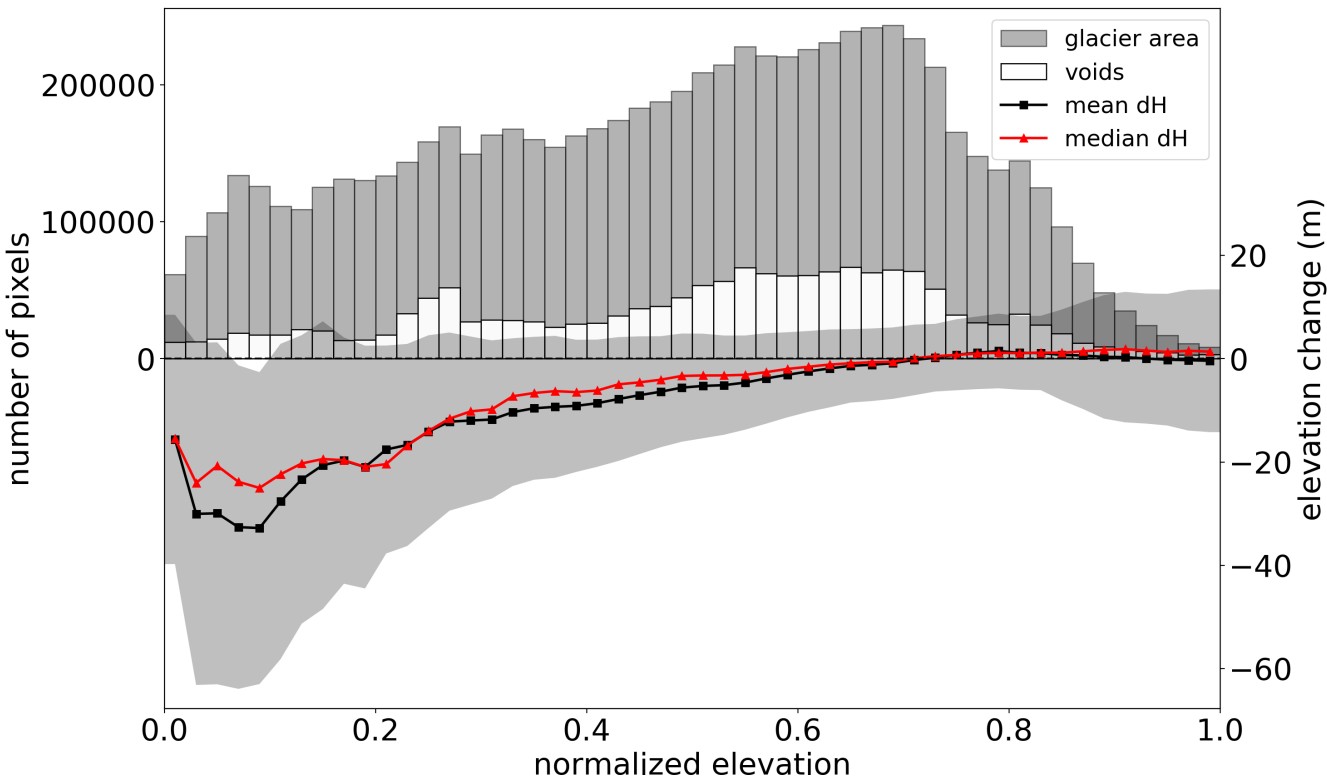

**Figure 3.** Distribution of glacier area and voids by normalized elevation (0 - lowest elevation, 1 - highest elevation), and mean and median elevation changes by normalized elevation, binned using bin widths of 0.02. The shaded area around the mean elevation changes indicates the mean $\pm$ one standard deviation.

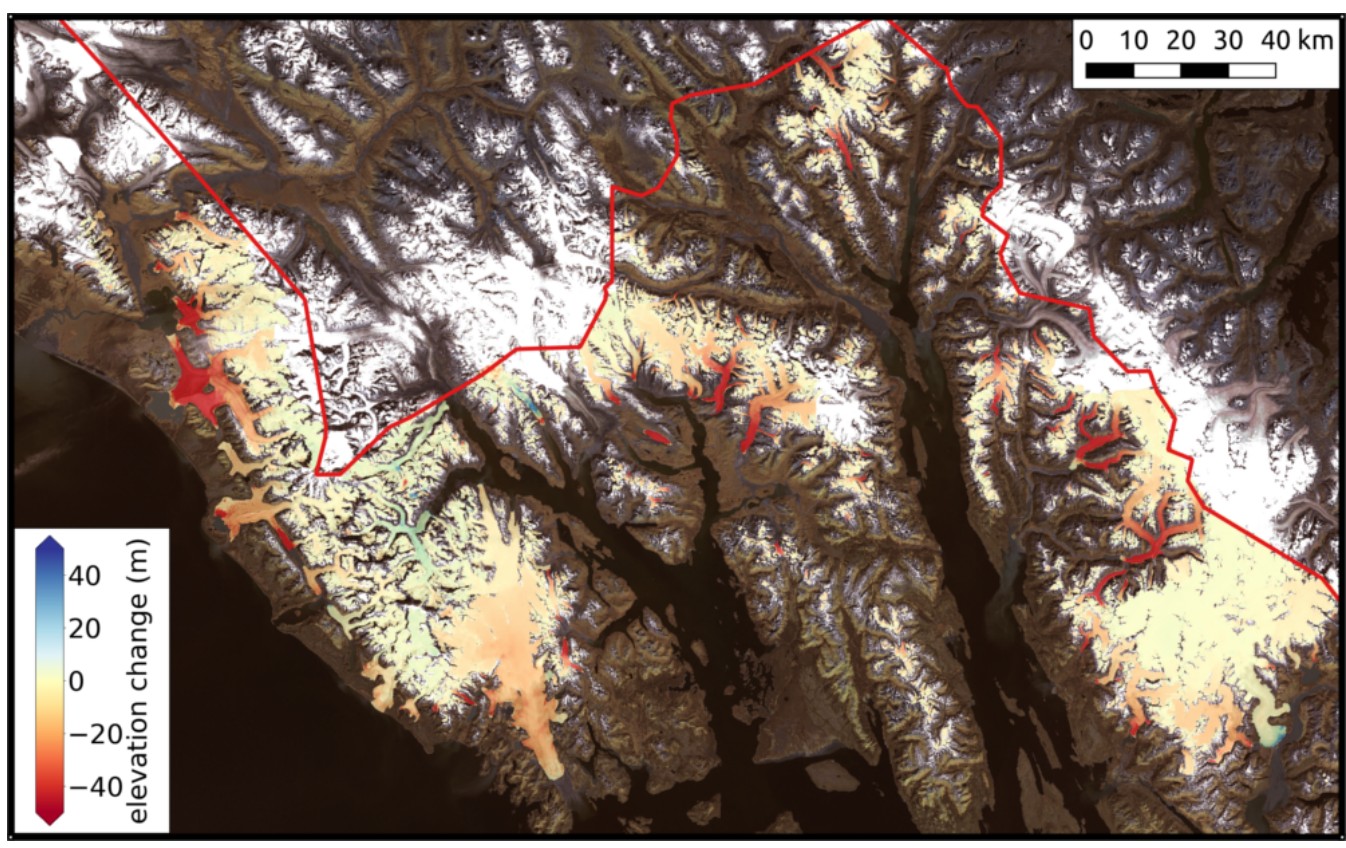

**Figure 4.** Non-voided, true elevation changes over the study area. Note contrasting patterns of thinning and thickening at lower elevations over glaciers labelled in Fig. 1., compared to the region in general. Background image is a mosaic of Landsat 7 scenes from 1999 and 2001; red line indicates location of US-Canada border.

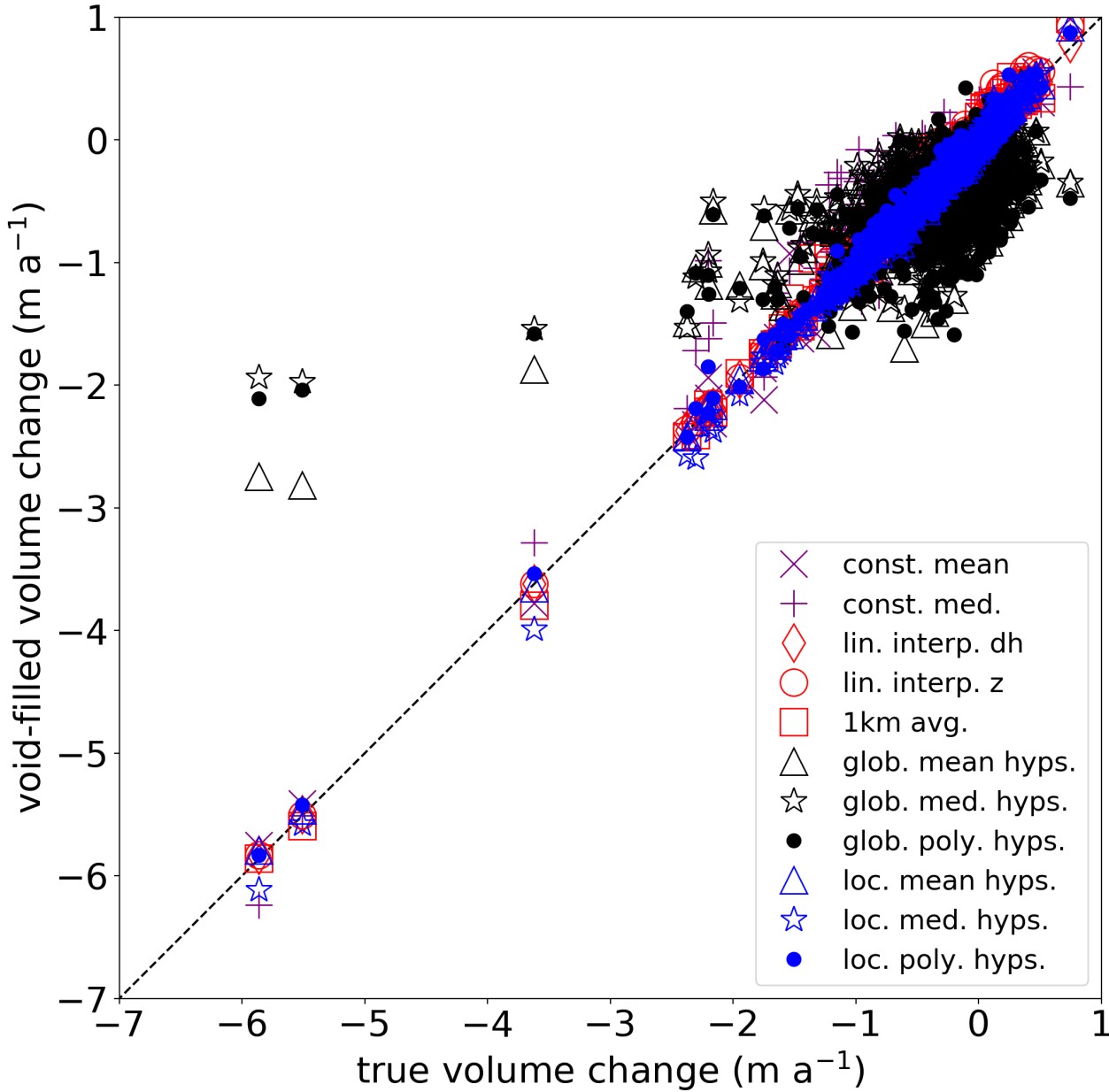

**Figure 5.** Void-filled volume change estimate for individual glaciers vs. true volume change.

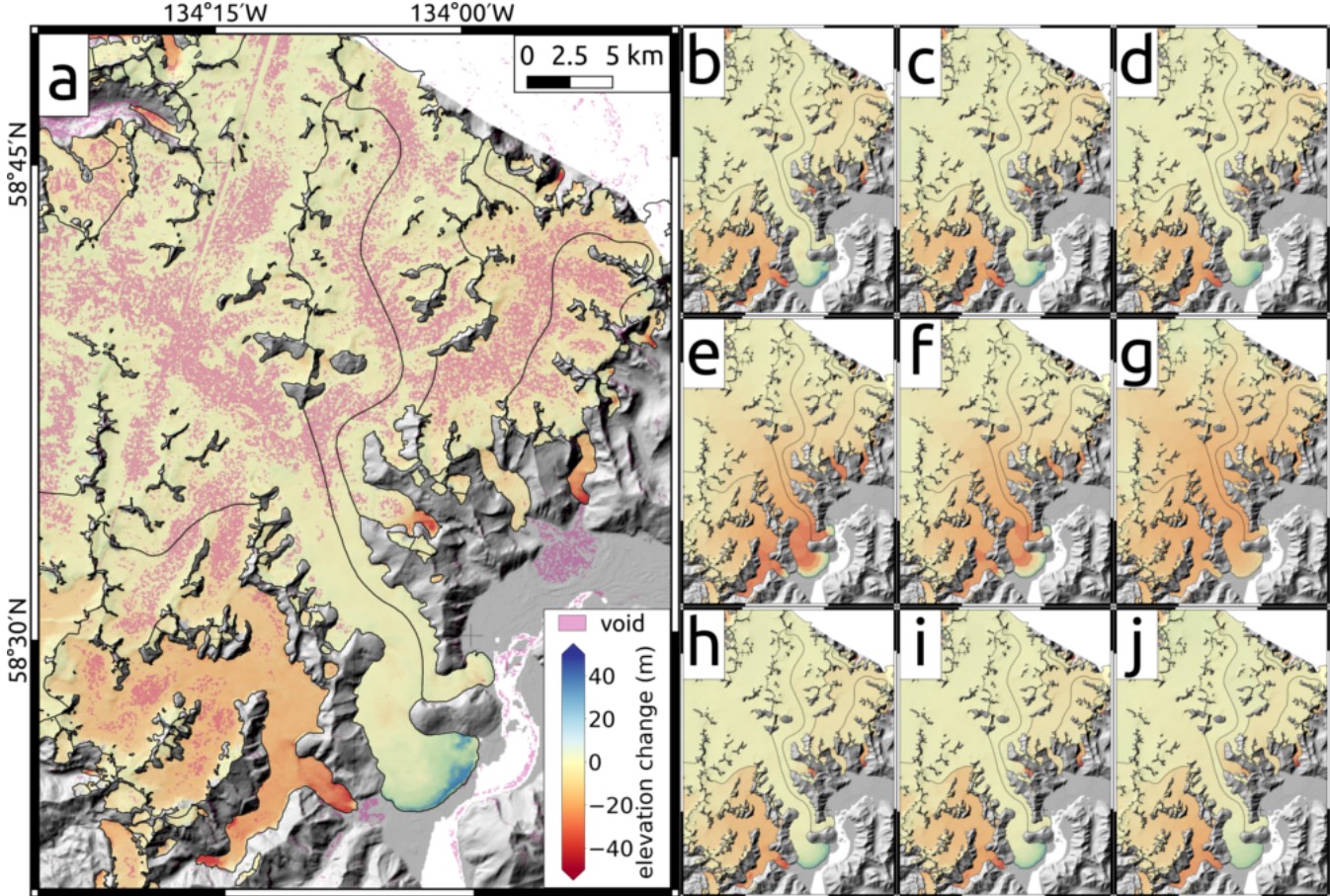

**Figure 6.** Elevation change maps for Juneau Icefield and Taku Glacier. (a) Initial, non-voided elevation change; (b) linear interpolation of elevation change; (c) linear interpolation of elevation; (d) 1 km average; (e) global mean hypsometric; (f) global median hypsometric; (g) global polynomial hypsometric; (h) local mean hypsometric; (i) local median hypsometric; (j) local polynomial hypsometric. Note that the global interpolation schemes in panels e-g show primarily surface lowering, in contrast to the actual signal of no change or surface increase, as well as increased notability of individual glacier outlines in panels h-j.

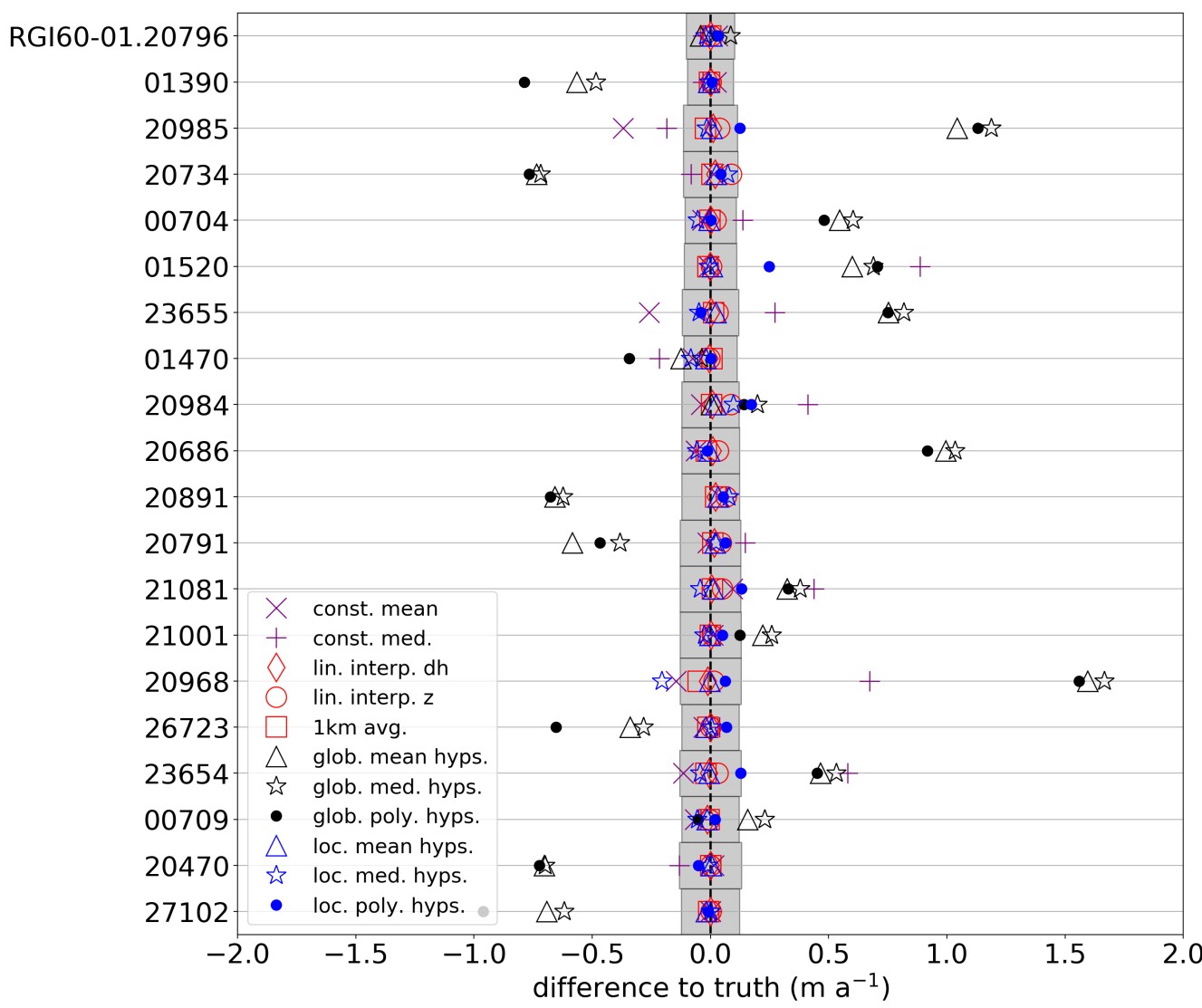

**Figure 7.** Comparison to true volume change for glaciers larger than $100\,km^2$, sorted by glacier area in descending order. Gray bars indicate uncertainty of the true volume change estimate for each glacier.

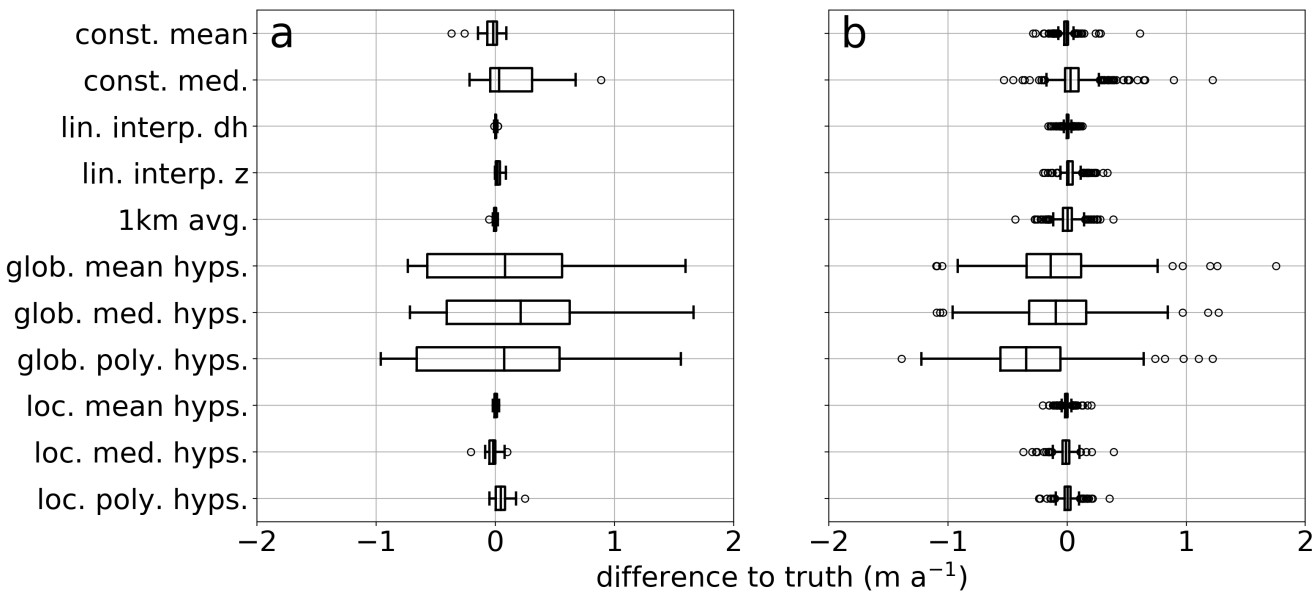

**Figure 8.** Difference to true volume change for each method tested, for (a) glaciers larger than $100\,\mathrm{km}^2$; (b) all glaciers.

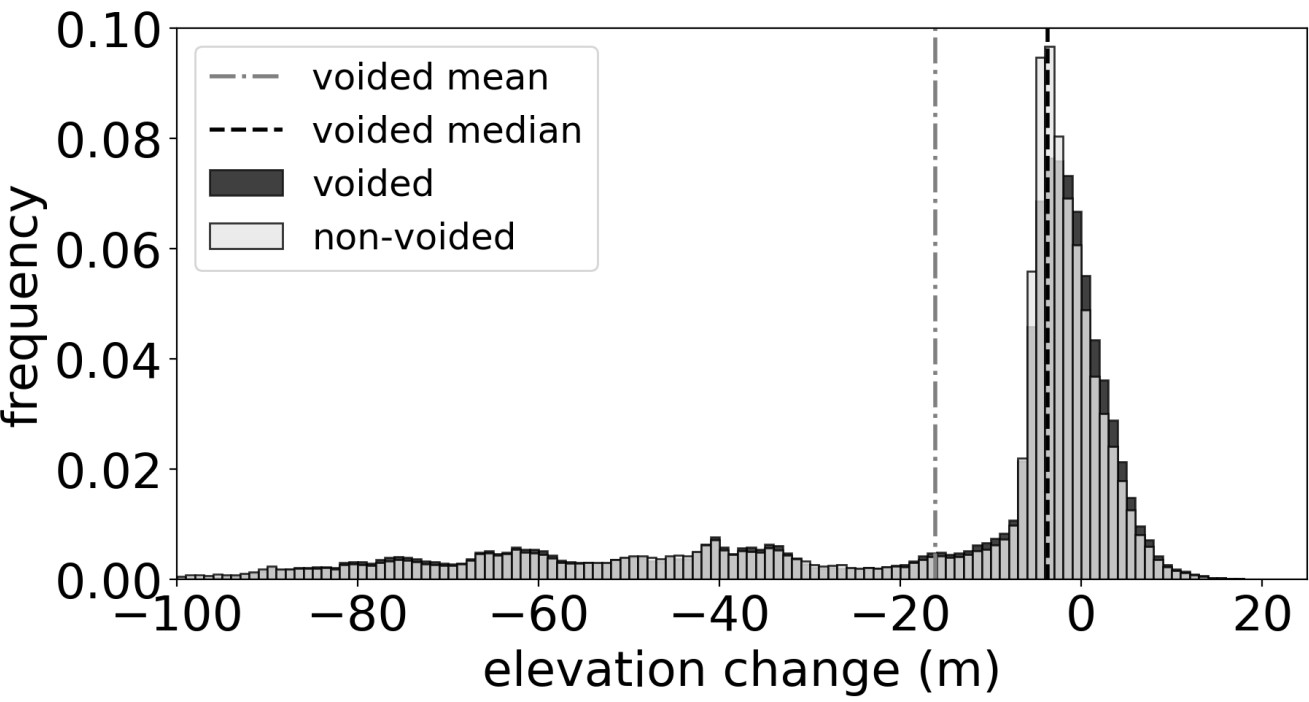

**Figure 9.** Distribution of elevation change values for voided and non-voided datasets over Field Glacier, binned using 1 m bins. Vertical lines indicate mean and median values for the voided dataset.

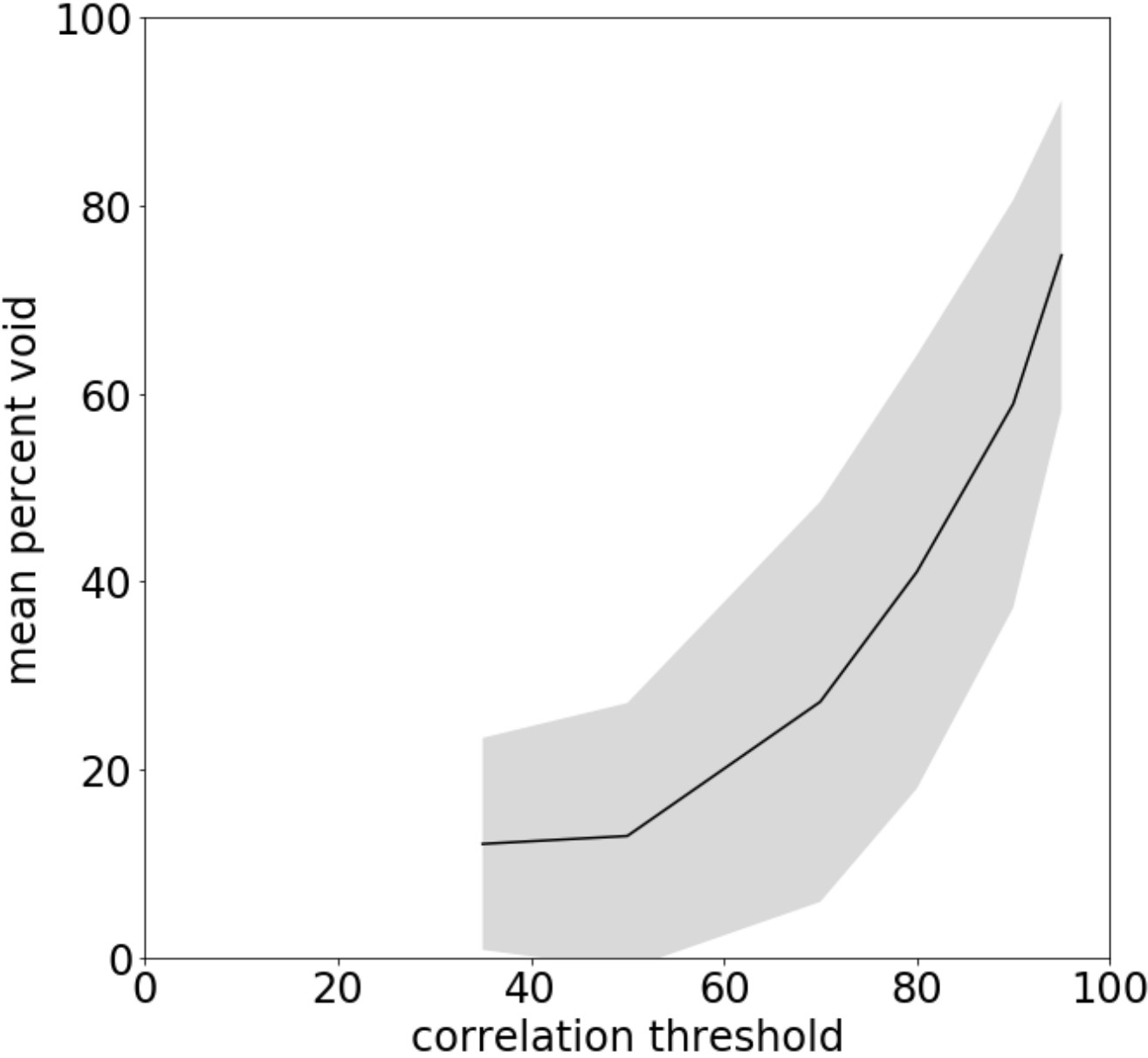

**Figure 10.** Mean (± standard deviation) percent void over glacier area for each of the correlation thresholds investigated.

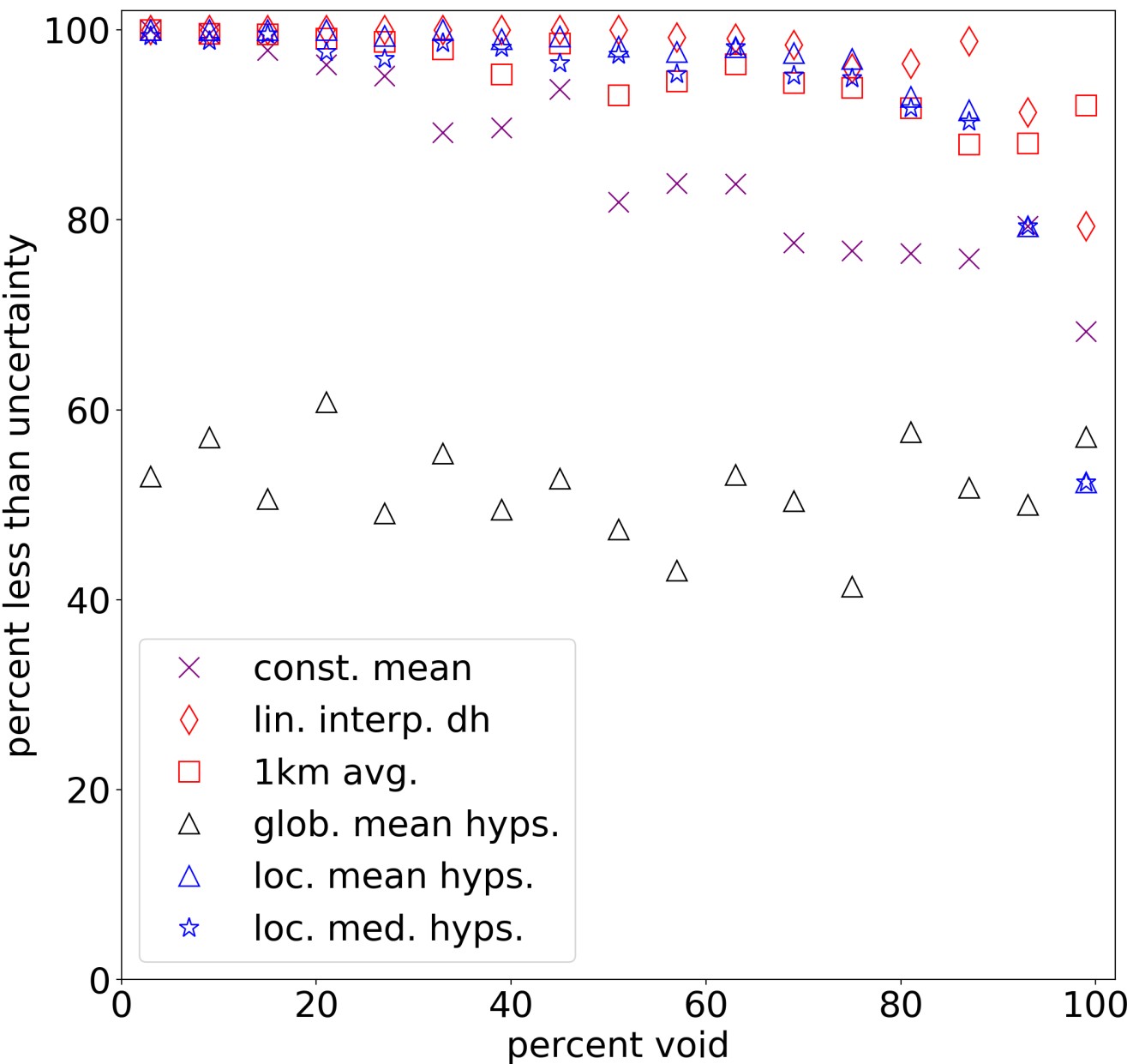

**Figure 11.** Percent of estimates that fall within the uncertainty estimates, as a function of the void percent.

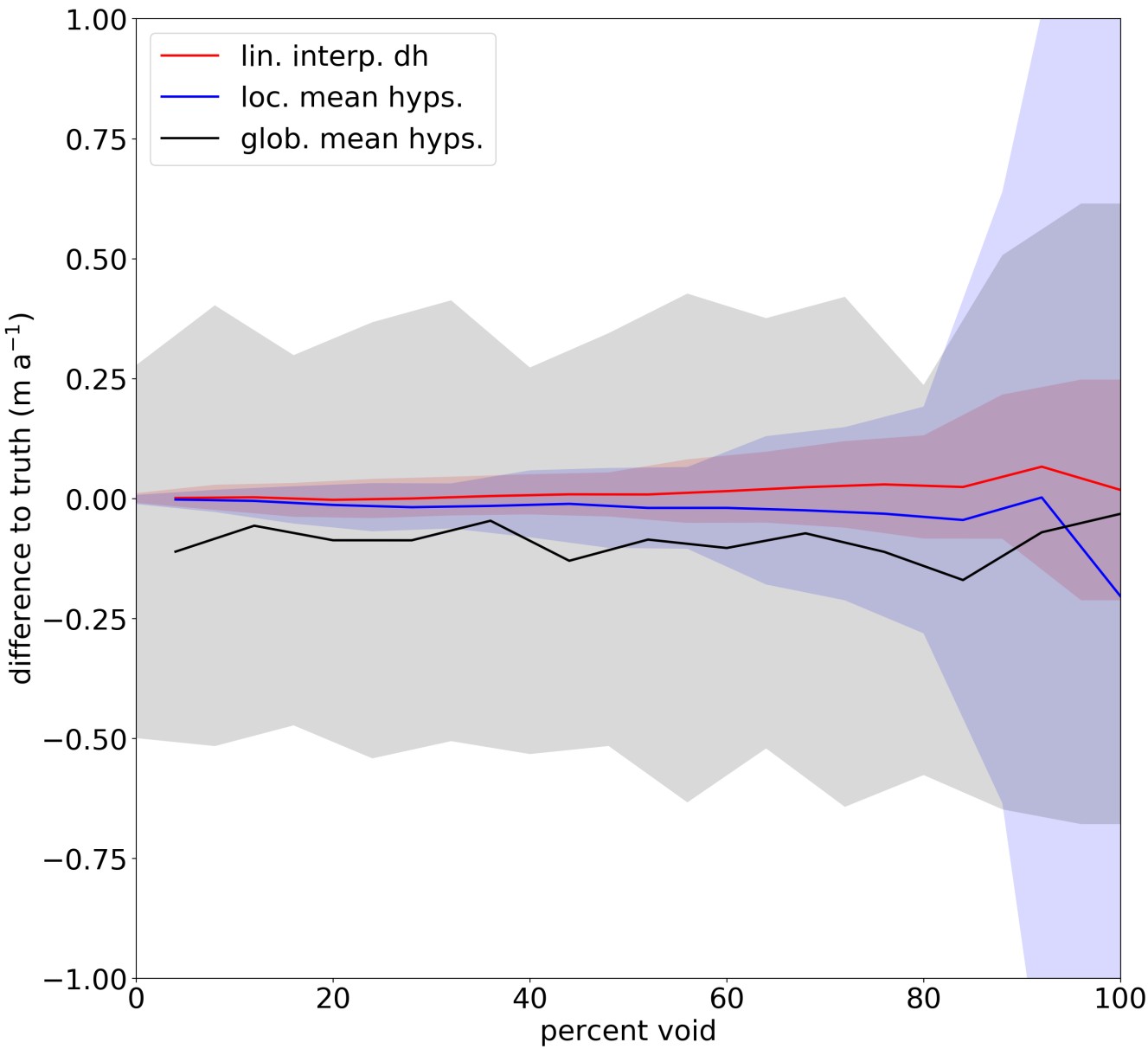

**Figure 12.** Difference to true mass balance as a function of void percentage, for the three best-performing methods on both an individual glacier and regional basis. Shaded region around each line indicates ± standard deviation.

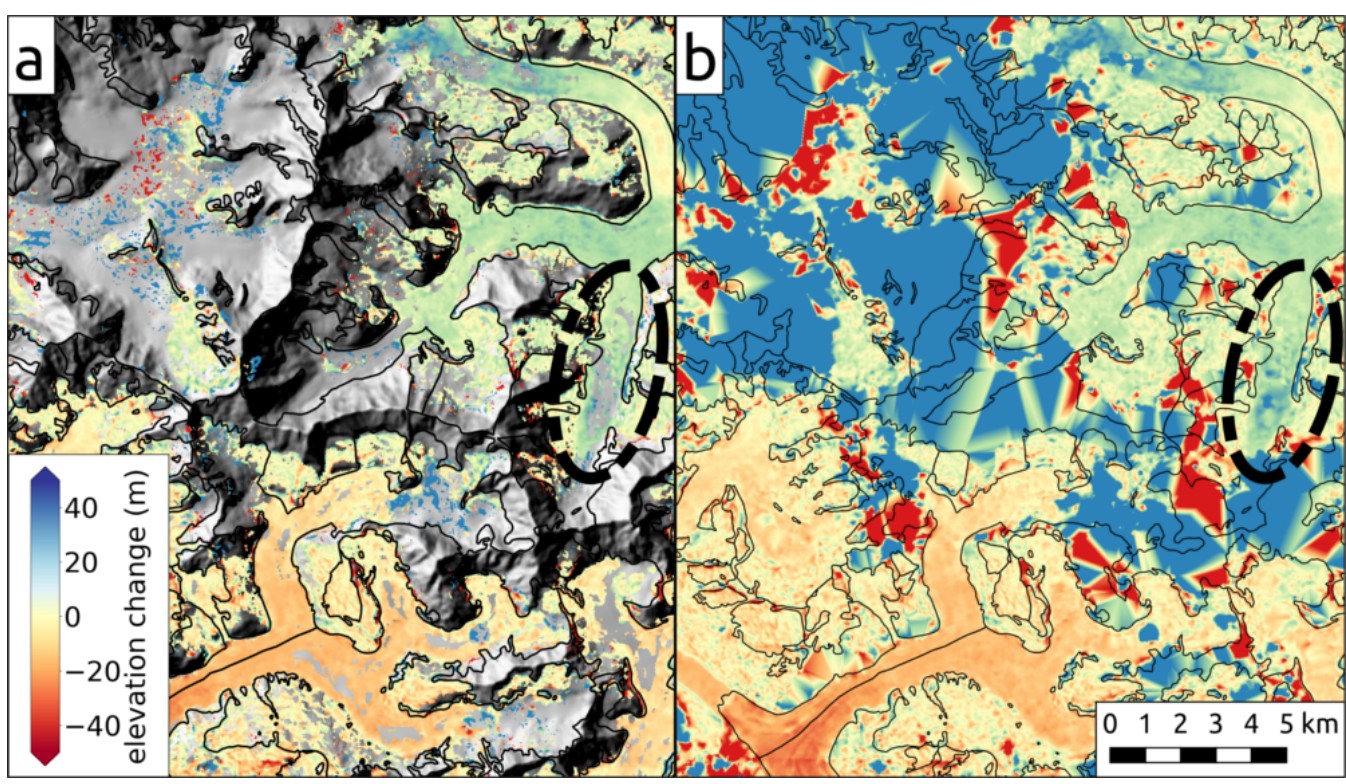

**Figure 13.** Elevation changes over the upper portion of Johns Hopkins Glacier, estimated by differencing an ASTER DEM acquired 13 August 2015 and the SRTM (a) with correlation-masked values left as nodata; (b) with voids filled using linear interpolation. Clear interpolation-related artefacts are seen in the sparsely-sampled accumulation area. Ellipses highlight an area over the glacier where linear interpolation performs well, with no obvious artefacts in the interpolated surface.

**Table 1.** Summary statistics for difference to true volume change for each method for the 415 glaciers sampled. Here, "std" refers to standard deviation, and "diff" refers to difference to true volume change (i.e., interpolated−true). All units in m a$^{-1}$, except for "total diff", which is in units of km$^3$ a$^{-1}$, and "pct. uncert.", which indicates the percentage of glaciers for which the interpolated dV was within the uncertainty of the true volume change.

| method | mean ± std | median | max | min | rms diff | total diff | pct. uncert. |
|---|---|---|---|---|---|---|---|
| const. mean | -0.01 ± 0.07 | 0.00 | 0.61 | -0.37 | 0.07 | -0.18 | 97.36 |
| const. med. | 0.07 ± 0.19 | 0.03 | 1.22 | -0.53 | 0.20 | 0.69 | 83.89 |
| lin. interp. dh | 0.00 ± 0.03 | 0.00 | 0.13 | -0.16 | 0.03 | 0.01 | 100.00 |
| lin. interp. z | 0.03 ± 0.06 | 0.01 | 0.34 | -0.20 | 0.07 | 0.16 | 99.52 |
| 1km avg. | 0.00 ± 0.08 | 0.00 | 0.39 | -0.44 | 0.08 | -0.01 | 99.28 |
| glob. mean hyps. | -0.08 ± 0.45 | -0.13 | 3.12 | -1.10 | 0.46 | 0.03 | 52.64 |
| glob. med. hyps. | -0.04 ± 0.49 | -0.08 | 3.92 | -1.09 | 0.49 | 0.52 | 51.92 |
| glob. poly. hyps. | -0.25 ± 0.51 | -0.32 | 3.75 | -1.39 | 0.57 | -0.39 | 37.02 |
| loc. mean hyps. | 0.00 ± 0.03 | 0.00 | 0.20 | -0.20 | 0.03 | 0.00 | 100.00 |
| loc. med. hyps. | -0.01 ± 0.06 | -0.01 | 0.39 | -0.37 | 0.07 | -0.09 | 98.80 |
| loc. poly. hyps. | 0.01 ± 0.06 | 0.00 | 0.35 | -0.24 | 0.06 | 0.26 | 97.84 |

**Table 2.** Difference to true volume change for two of the glaciers with the largest individual differences (Taku Glacier: 0.11 $\mathrm{m\,a}^{-1}$; Field Glacier: -1.15 $\mathrm{m\,a}^{-1}$). All units in $\mathrm{m\,a}^{-1}$, except for pct void.

| method | Taku Glacier | Field Glacier |
|---|---|---|
| const. mean | 0.03 | -0.01 |
| const. med. | -0.03 | 0.89 |
| lin. interp. dh | 0.00 | 0.00 |
| lin. interp. z | 0.00 | 0.00 |
| 1km avg. | -0.01 | -0.01 |
| glob. mean hyps. | -0.57 | 0.60 |
| glob. med. hyps. | -0.49 | 0.69 |
| glob. poly. hyps. | -0.79 | 0.71 |
| loc. mean hyps. | -0.01 | 0.01 |
| loc. med. hyps. | -0.01 | -0.01 |
| loc. poly. hyps. | 0.00 | 0.25 |
| pct void | 39.27 | 18.97 |