# Peer review of "Sensitivity of glacier volume change estimation to DEM void interpolation"

_The Cryosphere, 2018_

## Referee Comment (RC1) · N.E. Barrand (Referee) · 2 Oct 2018

Summary:

The authors provide a comprehensive assessment of the impact of void-filling routines on the calculation of glacier elevation and volume changes. This is an important work that has relevance for a wide variety of both local and regional scale glacier change studies utilising geodetic datasets. This is a timely study and a topic I've been interested in for some time. The manuscript is of high-quality, is very well written, largely free from errors, and suitable for publication in The Cryosphere. I would recommend acceptance following minor revisions, providing that the authors address the following minor comments. I'd like to congratulate the authors on an interesting study and an

important addition to the growing body of knowledge on regional-scale glacier volume change estimation. This paper will be an excellent companion to the equally good Nuth & Kaab TC study of 2010.

Minor comments:

- Title: There is an inconsistency between the use in the title of the term 'geodetic mass balance' and what is referred to elsewhere in the manuscript (and what is actually calculated) – which is volume change. I know why you have it up front in the title, as this is motivation for the study, but as you calculate only 'relative estimates of volume change' (4,23-24), the title is in fact incorrect. You do not assess the sensitivity of geodetic glacier mass balance in this work. The title therefore needs to be revised to 'volume change'. However, keep the geodetic mass balance mentions in the abstract and elsewhere, as they're used correctly there, and provide the important context to this work.

- page 1, line 18: can provide

- 1,21: has been calculated

- 1,24-25: this isn't quite right, though may just be a quirk of language. The geodetic method does not have to require extrapolation of sparse measurements, but it still can if measurements are sparse. Centreline elevation changes extrapolated to full width and differenced are still the 'geodetic method' (see, for example, Arendt references in your list). A couple of other studies, including one of mine, have directly compared mass balances calculated from full coverage DEMs and extrapolated centreline elevations (Barrand et al., 2010, J. Glaciol., 56, 199, doi:10.3189/002214310794457362).

- 2,4: not sure 'glacier water resources' is quite the phrase you're looking for as that gets into ice thickness / total water equivalent volume territory. Perhaps something like 'the scale of glacier change'?

- 2,35: I know you detail from where the DEMs are from later, but this sentence is

fragmentary and would benefit from a very brief description of the source of the data.

- 3,1-2: this sentence is strange. So, you're measuring volume changes but we should not interpret these as mass balance estimates? Why would we, given the additional density correction step that is necessary to calculate mass change? Why not calculate volume changes only (and present these) and avoid any mention of mass balance entirely? Then you solve the problem of seasonal timing. This looks to be what you've done (from the following sentence). If the estimates presented here '...should not be interpreted as mass balance estimates..', then you need to change the title of the paper and the content of the abstract, to reflect this.

- 17,1: it's not clear to me why the elevation data in this figure should be presented in a categorised colour scale. I think it would be clearer to view and interpret if the background hillshade was slightly opaque, and the DEM data were presented in a continuous colour scale. The dark grey outlines are presumably the ice-covered land, though this is not specified in the figure itself or the caption. With a more opaque hillshade, the ice cover would then be more discernable.

- 3,9-14: I don't think there is, but is there any reason to believe that findings from a single DEMs scene from this region would differ from elsewhere in the world (perhaps regional differences between SRTM tiles?). Can you justify here why this study uses just a single difference DEM from this location, rather than multiple difference DEMs from elsewhere?

- 3.20: qualify here that SRTM is commonly used at regional-scales and over medium to long time periods as it is not exceptionally accurate and likely wouldn't be as much use for e.g. 2000-2001 mass balances.

- 3,24-30: due to these problems, would it not have been better to select a region for which two high-quality regional-scale DEM products exist? Say, Iceland?

- 4,11-12: what's the justification for this omission now that we know that these very

small glaciers are quite important? (Bahr & Radic, 2012, Cryosphere, doi:10.5194/tc-6-763-2012).

- 5,1-2: specify 'most spaceborne stereo optical sensors'. Sensors onboard airborne platforms or historical aerial photographs will not have identical spectral range or resolution, and therefore may not be comparable with processing of ASTER scenes.

- 5,13: mean and median, or the mean or median? Which? See also 6,7-8.

- 5,20: if this is to be replicable then some more detail is required. Which surrounding pixels? Just those immediately proximal to the void? If so, this could be problematic as there may be inaccurate elevations just beyond the low correlation areas cutoffs. If not the very next pixel, then how many back from the void space? Provide enough detail of this method for another to reproduce your procedure exactly. See also 5,25

- 6,26: why 10%? What's your justification? 6,27: over what scales does spatial autocorrelation occur? I see this on the next page. But, why is it assumed to be 500 m (and why only 500 m given that it can occur on a range of scales simultaneously)?

- 18, Figure 2: Can you differentiate between the colour of the glacier outline and the ASTER correlation score mask? The middle panel all looks the same colour to me (except the red), even though I think its supposed to be dark grey outline and black mask.

- 19, Figure 3: Shaded grey around elevation changes refers to uncertainties? If so, please state in the caption.

- 7,9-10: why would you find the most voids occurring in the middle of the elevation range when from an optical image feature matching perspective (where the ASTER DEM gets its correlation score) you would expect fewer features and poorer correlation the higher up you go?

- 20, Figure 4: Background Landsat scene is a bit awkward to see as its so dark. Can you adjust the contrast, or similar to a previous comment, turn up the opacity to

de-emphasise the background and emphasise the elevations changes? Looks like a graded colour scale, yet legend shows categories. Shouldn't the legend by a graded colour bar too? Likewise other figures.

- 7,18: by acquisition area, do you mean accumulation area? If you're going to list individual glacier names in the main text, these need to be listed or shown in the figure somehow. 7,24: I would say 'patterns' isn't quite the right word here. Some of the 'variability' perhaps?

- 7,25-26: is it therefore worthwhile to consider repeating this exercise at the local glacier (rather than regional) scale? And for simple vs complex perimeter glaciers?

- 23, Figure 7: Great figure, but for readability perhaps the 'RGI60.01.' part can be removed from each individual glacier on the y axis and be included in a single y axis label? Can you also indicate in the figure caption how the individual glaciers are sorted along the y axis? It doesn't appear to be by RGI ID number, or by volume change. Is it north-south, or by glacier area, or something else?

- 24, Figure 8: It would be interesting to see this analysis extended to smaller glaciers, or the entire sample, but I understand if this is too time-consuming and therefore not possible.

- 9,1-20: some very small paragraphs here (comprising just one sentence sometimes). Is this necessary? 9,18-20: can you add some value judgments between these best three, perhaps quantifying precisely how each do and therefore which performs best? Actually, nevermind that, I see it in the next paragraph.

- 11,8: please quantify rather than just stating 'performed well'. 11,20-25: please replace 'do well', 'does well' etc, with 'perform(s) well'.

---

## Referee Comment (RC2) · Anonymous Referee #2 · 18 Oct 2018

Review of Mc Nabb et al., The Cryosphere, October 2018

Mc Nabb et al. compare different strategies of filling data gaps or interpolating sparse measurements of glacier elevation change in order to obtain the best estimate of total volume change (and ultimately glacier-wide mass balance). They assess the relative performance of the different gap-filling methods by comparing their results to the "true" volume change from the complete map of elevation change, an assessment both at the scale of individual glaciers and at the regional scale.

This is a certainly welcome study and I foresee that it is going to be widely cited. Indeed, almost all studies performing geodetic mass balance estimates need to handle data gaps. The procedure to assess the influence of different gap-filling method (i.e. taking a complete map of elevation change – dh – and creating realistic data voids in it) is adequate. That said, I was somewhat disappointed by the paper. It is not always clear and the writing could be improved. More importantly, I ended up with some questions that, I think, could have been, at least partly, answered. More work is needed to fully exploit this nice dataset and to transform this "good" study in a "benchmark" paper for the community.

**General comments.**

1/ Choice of unit to report the results. The authors have chosen to report their total volume change (and their departure from the "true" value) in $km^3$. I do not find this unit really useful, as it is so much dependent on the glacier area. This is why, most studies use the very convenient unit of m w.e. $yr^{-1}$ (or kg $m^{-2}$ $yr^{-1}$) to report mass balances. With the latter unit, it is easy to compare different glaciers within a region or glacier mass balance from different regions. I fully understand (and support the fact) that the authors do not want to provide mass balances here because many additional corrections would be required to obtain a meaningful value. Thus, I suggest that they use glacier-wide or region-wide elevation change (thus in meters), together with % of error (as already done).

2/ How to handle data gaps in the error estimate. A missing section/discussion is how to take into account the data gaps in the formal error estimate. Right now, authors performed a sound sensitivity analysis and conclude on the best strategies, which is already useful. However, a remaining question is how to include the uncertainties dues to data gaps in the formal error estimate. I do not think this is done well in the literature so far and I was hoping to find an answer here. Authors would increase the impact of their work if they could provide, at least, some suggestions. I know this is not straightforward but really hope they can tackle this issue.

3/ % of data gaps. The gap creating method makes sense. However, I had the feeling that the % of data gaps was not very high and the data voids not large. Are these percentages of data gaps in line with published values? A more aggressive gap creating threshold is discussed, but too briefly. How much data gaps are created in this case? I think many readers would be curious to know if the conclusions hold when ~50 % (or more) of data gaps are present.

4/ Variability of dh in the study region. I miss a more thorough description of this variability. This is important here because in an end-member case (hypothetic) where there would be no spatial variability of elevation change, then most gap-filling methods would work well. How does variability vary with elevation? I expect less dh variability at high elevations where data gaps tend to be concentrated, which may explain why the local hypsometric approach works well. To quantify variability, individual glacier mean elevation change (not glacier-wide mass balance) could be calculated and the spread shown. How does this spread compare to earlier studies? It would also help to discuss whether the study region is representative.

5/ Global hypsometric approach, normalized elevations or not? To take into account the diversity of the altitude range of glaciers in a region, some earlier studies have normalized the elevation in order to extrapolate to un-surveyed areas. This is also what the authors do here to plot dh in their Figure 3. I was wondering if the normalization helped or not for the extrapolation. This procedure seems to make sense and it would be good to test its added value.

**Specific comments.**

1.1 "mass balance" does not "imply sea level". Glacier mass gain/loss does.

1.2 Mentioning glaciological measurements in the abstract is not really useful. Not the core of the paper.

1.5. Is "based" the best word here?

1.18. One further and strong important limitation of the glaciological mass balances is that they seem to be performed on glaciers where the mass balances tend to be more negative than the regional average (Gardner et al., Science, 2013).

1.20 They must be a reference for the WGMS data and also for the number of glaciers on Earth

1.22 A reference to a review? Possibilities I see are:
Bamber, J. L. and Rivera, A.: A review of remote sensing methods for glacier mass balance determination, Global and Planetary Change, 59(1–4), 138–148, doi:10.1016/j.gloplacha.2006.11.031, 2007.
Bamber, J. L., Westaway, R. M., Marzeion, B. and Wouters, B.: The land ice contribution to sea level during the satellite era, Environmental Research Letters, 13(6), 063008, 2018.
Marzeion, B., Champollion, N., Haeberli, W., Langley, K., Leclercq, P. and Paul, F.: Observation-Based Estimates of Global Glacier Mass Change and Its Contribution to Sea-Level Change, Surveys in Geophysics, 38(1), 105–130, doi:10.1007/s10712-016-9394-y, 2017.

1.25. Do the authors exclude from the geodetic method (and thus from the study) all ICESat-based estimates of glacier volume change? ICEsat provides sparse measurements that need extrapolation. To be clarified.

2.6. Acronym "DEMs" to be used here, as defined already.
Do the authors understate that they exclude estimate based on ICESat or sparse GPS surveys?

2.9 Maybe a short statement that this is certainly true for old imagery (8-bits) but that this issue is strongly reduced using state-of-the-art 11- or 12-bits stereo data? In the end, I also note that the data gaps are not so concentrated in the accumulation area.

2.25 I think the interpolation methods should be described only once but not "briefly". They are the heart of the study.

2.28. Does it make a difference that the elevation with altitude is used to fill unsurveyed values vs. just multiplied by the area of the altitude band? For the glacier-wide mass balance (or the glacier-wide dh) I think it is the same. Maybe state it to avoid confusion for some readers.

2.30 I very strongly suggest using "regional" instead of "global". I found "global" confusing (I immediately thought about the whole Earth). Or did I miss a difficulty linked to the use of "regional"?

2.31 "basin" needed after "glacier"?

2.35 The sentence "In this paper, we use two high-quality, radar-derived DEMs." does not appear to be complete and break the flow of the introduction.

3.14. I think the key point for this study is that the authors have a large intra-glacier and inter-glacier variability of elevation change (a consequence of the variety of glacier type). Make it clear and quantify better (see general comments). The authors may note that some previous workers have separated different glacier types while extrapolating.

3.24. % of data gaps in SRTM for this study area?

4.26 Could also have been done on the SRTM. Maybe state that this is an arbitrary choice.

5.3 How did the authors handle clouds in ASTER?

5.9 As said before, description of each interpolation method is central to the study. So we do not want to have a "brief summary" only. In fact the description is detailed enough.

5.14 Here and elsewhere I found the use of "glacier basin" instead of "glacier" a bit problematic. For me a glacier basin includes the glacier + the off-glacier terrain included in this basin. Why not using "glacier" simply ? (everywhere)

5.22 "linear interpolation". Should not it be "bilinear"?

5.22 "because the voids are relatively small" is not a very precise statement. It lack quantification (void size?) and one also would like this study to address the case of large data voids.

6.8 is 'original elevation' clear enough?

6.27. IMPORTANT. I see no reason why the systematic error in elevation difference (epsilon_bias) obtained using triangulation between the DEMs should be divided by the square root of the number of effectively independent pixels. Either justify or correct.

7.8 I would have expected a higher percentage of voids in the accumulation area. This is not the case. This should be discussed.

7.14 title of section 4.2 is not really meaningful. Improve section and sub-section titles if possible.

7.16. An elevation change can be negative, not a pattern.

7.21 " The pattern of elevation change shown on Rendu Glacier in the elevation difference maps". Authors need to improve the text.

7.26 to 8.2. These sentences are not really well written and the reasoning is hard to follow. In fact, I do not see the rational for using volume change in $km^3$ (and quoting an average volume change). This unit is so much dependent on the size of the glaciers whereas the global hypso method consist (if I understood correctly) in using mean/median dh per elevation band. So if a glacier (whatever his size) as a dh vs. altitude pattern like the rest of the region then the method should work.

8.4 the fact that the authors do the conversion here to average elevation change (in meter), nicely illustrates the limit of the total volume approach (in km3).

8.7. IMPORTANT. The fact that the authors interpolate "over much smaller areas" (and the authors are aware of that) is quite problematic. It suggests that the authors are in a configuration (with sparse data voids) where local gap filling methods will all perform reasonably well. A much more aggressive gap creating strategy should be considered in an alternative scenario.

8.8-10. I do not follow the reasoning. Contour lines are maybe (certainly) biased at high elevation but a DEM created from them does not have data gaps. So the fact that contour line is floating is a different problem (like radar penetration) and does not influence the errors due to gap filling. Or better explain if I missed something.

8.30 Showing the dh with altitude for each of these 20 glaciers and the regional mean value would nicely illustrate the text.

9.11 Did authors used the term "global fits" before. I do not think so. If they want the readers to follow them, then they need to stick to a terminology.

9.22 "Differences" of what?

9.26 authors need to clarify "relative". Is it normalized? If yes, I think they should quantify the added value of the normalization for the same global mean hypsometri method.

9.30 "one explanation for the value". Do the authors want to discuss a high/low "value"? Clarify. Did they expect this method to perform better? Avoid such understatements.

10.9 do the authors suggest using the median rather than the mean as a metric of centrality for an elevation bin? I think it could be dangerous because the dh distribution could also be quite skewed with an elevation bin (when it comes to large glaciers for example). At least this needs to be discussed.

10.11 Authors need to provide the corresponding % of data gaps? Does this more aggressive threshold really lead to a strong increase in data gaps? Where on the glacier?

10.22 Authors should detail how the ASTER DEMs were derived. Depending on the methods (and correlation threshold) the percentage of data gaps will change quite a lot. The following question is thus raised: Is it better to keep only the most reliable values in the DEM and increase data gaps (and filled them afterward) or alternatively try to get the DEM processing parameters resulting in the most complete DEM. If the authors could also contribute to this research question they would increase the impact of their study.

10.25 was dDEM defined already? (not sure)

10.29 Is this value of "0 km³" the volume change estimate, suggesting surprisingly no volume change? Or the difference to the "true" IfSAR/SRTM value?

10.31 "3.6" positive value of volume change? OK?

11.1 surprising statement that the two methods perform as well when authors just illustrated the danger of the linear interpolation method…

11.25 the dependence on the size of the voids has unfortunately not been examined sufficiently.

11.30 I do not think this issue of proximity has been really addressed so that such a conclusion can be made. Or I misunderstood the statement? Do the authors suggest using a modified global method using only the glaciers in the vicinity of the one for which volume change needs to be calculated?

11.33 "suffice" well anyway there is no other choice right? If only a few "anomalous" glaciers are sampled than the regional total could be strongly biased

Table 1. Can the authors tell if these are simple (as I guess) or area-weighted statistics? Maybe remind in the legend the number of individual glaciers on which these statistics are obtained.

Figure 1. I could not find name of glaciers on this figure.

Figure 3. What is the envelop around the mean/median dh? 1-sigma of data?

Figure 2. An extra panel showing the distribution of data gaps for the more aggressive correlation threshold would be welcome. Also provide on each panel the % of data gaps for Taku Glacier.

Figure 5. The authors use "Actual volume change" here but "true volume change" in the text. Homogenize. Are all the acronyms used to name the different methods in the figure defined (in the text or the legend)?

Figure 6. Rather than showing the dh maps for all methods (with some maps that are very similar), it would probably be best to show only the ones with strong difference. Also it would be good to show the map with data voids. So that the reader as a good sense of where the voids where.
Authors could also consider moving this figure (or the suggested revised version of it) to the supplement. Showing instead the pattern of change with altitude for Taku derived from these maps could likely better illustrate some of the subtle differences mentioned in the text.

---

## Referee Comment (RC3) · Anonymous Referee #3 · 19 Oct 2018

In this paper, McNabb et al. investigate the effect of missing data (called "voids" in the article) on the glacier volume change that can be obtained from digital elevation models (DEMs) differencing. The methodology is rather straightforward, they differentiate two DEMs acquired over Southeast Alaska that (almost) cover the entire glacierized area. These data are used as reference data. Then, they artificially generate voids in the data and evaluate the impact of different void-filling/interpolation methods on the on the regional glacier volume change estimate, but also for each individual glacier. They investigate 11 different void-filling/interpolation methods that are often used in the literature, providing a unique and comprehensive assessment. They conclude that most interpolation methods introduce very little bias (<1%) on the regional glacier volume change. However, individual glacier volume change estimates can be severely

affected by the choice of the interpolation strategy.

This paper is rather narrow-focused, but its scope fits very well within The Cryosphere, where it will certainly reach an adequate audience. The topic is timely and very relevant, as the geodetic method is more and more widespread in glaciology. To my opinion, this paper has the potential to become a classic paper in the field of geodetic mass balance. However, and while I appreciate the concision of the paper, I have the feeling that the authors could discuss some aspects more in depth. Moreover, I sometimes had a hard time following the paper and found that it lacks clarity in its current form. These are my two major comments.

Major comments:

1-Volume change vs. geodetic mass balance

The title of the paper mention the sensitivity of "geodetic glacier mass balance", but actually discuss only glacier volume changes. This decision is somehow understandable, because it is the quantity that is directly affected by the void-filling strategy. However, the impact of void-filling strategies on the individual glacier volume change expressed in km3 is not very intuitive, and not as informative as it could be. First of all, the IfSAR DEM was acquired over two years and it would be better to present the annual mean instead of the totals, in order to get rid of this temporal inconsistency. Second, the results are largely dependent on the glacier area considered, larger glaciers being more sensitive to the interpolation (P8L12-13), mostly because for a similar elevation change they have larger volume change, due to their larger area. For example, for figures 7 and 8 (and 5?), I suggest to present the results in kg m$^{-2}$ a$^{-1}$ or in m a$^{-1}$ (if the authors do not want to make any density assumption).

If the authors want their study to be reproduced and the conclusions of this article to be applied elsewhere, they need to analyze the influence of the void-filling strategy more in depth. I feel like the paper misses some basic, yet interesting analysis. For instance, what is the influence of the percentage of voids for individual glaciers? Of the glacier-wide mass balance/mean rate of elevation change? Of the glacier area? The authors probably analyzed these influences already and found that they were limited/not interesting, but I think it is probably worth mentioning them, in order to apply their conclusions to a different setting.

2-Some clarifications needed

The objective of the study is quite straightforward, but a number of confusions and unclear statements prevent from an easy understanding of the paper. I had to go back and forth a number of time reading the paper, and I have the feeling that the clarity of the paper could be much improved if the authors address the three comments below.

First, the author mimic the voids of a standard DEM difference, based on ASTER correlation map patterns. Consequently, I expected that they would investigate the influence of the void-filling strategy for this purpose. However, they also investigate such methods as the "global" ones, which are generally used for regionalization of Lidar surveys. They should make a clear distinction between these two applications when relevant. In other words, I do not think that the "global" methods are relevant for DEM differences void filling at the scale of individual glaciers. Correct me if I'm wrong, but I do not know any paper which studied individual glacier mass balances obtained with such "global" methods to fill in the holes of a DEM difference.

Second, the different methods described are relatively basic, however their description should be clearer. For instance, adding equations to the description of each method would be beneficial. Alternatively, you could share the code you wrote, which would also support your conclusion in which you encourage others to test different methods when dealing with voided data.

Third, I found the example about Taku Glacier extremely confusing. If I understand correctly figure 2, the tongue of Taku Glacier is mostly free of voids. However, the global methods (panels e to g) totally change the pattern of areas where data are available! Consequently, the methods should be described as "Interpolation" and not "Void-filling" (for instance the title of section 3.2 should be changed), because they also apply to areas without voids (and it is technically a non-exact interpolation method). If I did not understand correctly figure 2, you can ignore this comment, but you should consider changing figure 2.

Specific comments:
Fig. 2: confusion between the glacier and voids outlines. You should draw the glacier outlines in a different color/line thickness, such as the panel 3 is easier to understand.

Fig. 5: add a scale/grid on the inset.

Fig. 6: this figure is extremely confusing to me. First of all, I'm missing the location of the voids on panel a, and I had to go back on the previous figures to understand where the voids where. Then the figure shows the strong impact of the void-filling procedure on the tongue of Taku Glacier, but from what I understood of figure 2, there were no voids on the tongue. This comment is in line with my major comment 2.

Fig. 7: I think here the reader loses the information about the difference to truth relative to the total glacier volume change. I guess the larger the volume, the larger the error.

P1L3 and P1L17: I do not fully agree with your definition of the glaciological method, which does not really monitor changes in surface height (that would actually be the geodetic method). The glaciological method directly measures the surface accumulation and melt.

P1L8-9: add a word about the artificially generated voids

P1L11: define ASTER

P2L6: "Digital Elevation Models"-> DEMs

P2L33-P3L6: this paragraph is not really well structured in my opinion. You should describe more clearly the philosophy of your study. I suggest to move completely your warning statement about the radar DEM difference (P3L1-2) to the other place where it is mentioned (P4L19-24). You need to add something about the artificially generated voids and to better justify the choice of ASTER, among a large choice of (optical) sensors.

P3L9-14: what is the glacierized area?

P3L19-20: provide references about studies that used SRTM to estimate geodetic glacier mass balance

P3L29: what proportion of the glacierized are if affected by these "small" voids?

P4L2-6: more references and details are needed in this paragraph. What is the precision of the IfSAR DEM? What is the proportion of voids? Has it been used in other glaciological studies?

P4L5-6: is this sentence useful?

P4L12: can you justify the exclusion of glaciers smaller than 1 km$^2$ from your analysis? Method section: at some point you need to explain how you calculate the regional estimate. Is it the sum of the individual glaciers, or do you consider all the glaciers as a single body of ice? Do you include the glaciers smaller than 1 km$^2$ in this estimate?

If not you might bias it.

P5L9: see my major comment 2 -> you might want to rename this paragraph "Interpolation" instead of "Void Filling"

Void-filling section: how do you deal with the temporal inconsistency between your two IfSAR DEMs? I guess for most method you interpolate the rate of elevation change and not the elevation change? This should be written clearly. However, this is not possible for the method based on the interpolation of elevation.

P6L4-6: here the reader wonders why using ASTER voids for regionalization applications (i.e., Lidar based studies)? In order to test the influence of regionalization, one could extract elevation changes in your DEM difference along Lidar flight lines... but this would be another paper!

P7L6: define "normalized glacier elevation"

P7L14: consider switching the order between the sections 4.2 and 4.3.

P7L15-23: this paragraph is rather disconnected from the rest of the analysis.

P7L16: "the pattern of elevation change is negative" -> the phrasing is not clear to me

P7L26-27: the average geometric volume change has probably little influence, contrary to the mean elevation difference.

P8L3-10: can you say a word about the constant methods? And about the 1km neighborhood method?

P8L4: define RMS. This sentence is not completely clear to me.

P8L13-14: the larger glaciers are more sensitive than the others, because they have larger volume change for a similar elevation change, due to their larger area. This is one on the limitations of your analysis, because you look at volume changes only (see my major comment 1). You should consider extending your analysis to glacier mass

balance, or rate of elevation change.

P9L6-8: this paragraph is very short, while Fig. 8 is probably a key figure! Could you elaborate a bit? I found an unbalance with the previous paragraph (P8L30-P9L5), which is less important and much longer than this one.

P9L20: the 1km neighborhood is never mentioned earlier in the section 4.2.

Section 4.3: consider adding a column to Table 1, which summarizes the regional totals for each method. It might also be interesting to discuss the difference between the regional estimates obtained by summing the individual glaciers versus the regional totals obtained by applying each method to the glacierized area considered as a single body of ice.

Section 4.4: why don't you use the percentage of voids of individual glaciers to study the influence of the percentage of voids on the distance to truth?

P10L12: give the total percentage of voids for each threshold.

P10L28: again I got confused because you look at 91 individual glaciers and then you mention the "global mean hypsometric" as one of the best performing method... Please clarify.

P10L31: you can mention in the text that outliers are more often located near the voids, which increases their influence in a linear interpolation.

P11L5-7: the order (regional volume change then individual glacier volume) is the opposite from section 4.2 and 4.3.

P11L25-26: you actually did not demonstrate this in your analysis...

---

## Author Comment (AC1) · 25 Jan 2019

**Reponse to Reviewer 1**

**1 Summary**

The authors provide a comprehensive assessment of the impact of void-filling routines on the calculation of glacier elevation and volume changes. This is an important work that has relevance for a wide variety of both local and regional scale glacier change studies utilising geodetic datasets. This is a timely study and a topic I've been interested in for some time. The manuscript is of high-quality, is very well written, largely free from errors, and suitable for publication in The Cryosphere. I would recommend acceptance following minor revisions, providing that the authors address the following minor comments. I'd like to congratulate the authors on an interesting study and an important addition to the growing body of knowledge on regional-scale glacier volume change estimation. This paper will be an excellent companion to the equally good Nuth & Kaab TC study of 2010.

We would like to thank the referee for their careful and constructive comments that have helped to improve both the clarity and focus of the manuscript. Our responses to the comments below are in blue, with the original comments in black.

**2 Minor comments**

- Title: There is an inconsistency between the use in the title of the term geodetic mass balance' and what is referred to elsewhere in the manuscript (and what is actually calculated)  which is volume change. I know why you have it up front in the title, as this is motivation for the study, but as you calculate only relative estimates of volume change' (4,23-24), the title is in fact incorrect. You do not assess the sensitivity of geodetic glacier mass balance in this work. The title therefore needs to be revised to volume change'. However, keep the geodetic mass balance mentions in the abstract and elsewhere, as they're used correctly there, and provide the important context to this work.

  We agree, and have changed the title to better match the text.

- page 1, line 18: can provide

  Changed.

- 1,21: has been calculated

  Changed.

- 1,24-25: this isn't quite right, though may just be a quirk of language. The geodetic method does not have to require extrapolation of sparse measurements, but it still can if measurements are sparse. Centreline elevation changes extrapolated to full width and differenced are still the geodetic method' (see, for example, Arendt references in your list). A couple of other studies, including one of mine, have directly compared mass balances calculated from full coverage DEMs and extrapolated centreline elevations (Barrand et al., 2010, J. Glaciol., 56, 199, doi:10.3189/002214310794457362).

  Added a parenthetical statement to make clear that we aren't excluding, for example, laser altimetry/ICESat studies from the 'geodetic' label.

- 2,4: not sure glacier water resources' is quite the phrase you're looking for as that gets into ice thickness / total water equivalent volume territory. Perhaps something like the scale of glacier change'?

  Changed to 'scale of glacier change.'

- 2,35: I know you detail from where the DEMs are from later, but this sentence is fragmentary and would benefit from a very brief description of the source of the data.

  See response to next comment.

- 3,1-2: this sentence is strange. So, you're measuring volume changes but we should not interpret these as mass balance estimates? Why would we, given the additional density correction step that is necessary to calculate mass change? Why not calculate volume changes only (and present these) and avoid any mention of mass balance entirely? Then you solve the problem of seasonal timing. This looks to be what you've done (from the following sentence). If the estimates presented here . . .should not be interpreted as mass balance estimates..', then you need to change the title of the paper and the content of the abstract, to reflect this.

  We have moved this sentence to the end of the paragraph, and included information about where the DEMs come from (C-band vs. X-band). We have also made it clear that additional corrections (density, seasonal timing) must be made before these values are interpreted as mass balances. Additionally, we have made it more clear that we are looking at the effects on volume changes, which can then be used to estimate mass balances, in response to your previous comment.

- 17,1: it's not clear to me why the elevation data in this figure should be presented in a categorised colour scale. I think it would be clearer to view and interpret if the background hillshade was slightly opaque, and the DEM data were presented in a continuous colour scale. The dark grey outlines are presumably the ice-covered land, though this is not specified in the figure itself or the caption. With a more opaque hillshade, the ice cover would then be more discernable.

  The color scale is continuous, but QGIS displays the legend as a non-continuous scale. We have added a continuous color scale to the legend, specified what the dark outlines are, and increased the transparency of the background hillshade.

- 3,9-14: I don't think there is, but is there any reason to believe that findings from a single DEMs scene from this region would differ from elsewhere in the world (perhaps regional differences between SRTM tiles?). Can you justify here why this study uses just a single difference DEM from this location, rather than multiple difference DEMs from elsewhere?

  We don't believe that there would be a significant difference in the results from this region vs. another region, in part because of the diversity of glacier types, sizes, etc. that are found in this region. The reason to choose a single DEM difference is that then the results are not dependent upon variations in the changes through time. In this respect, the effects of void interpolation is most easily extracted and understood using a single DEM difference. By using a large collection of varying glaciers in one region, we also simulate something similar to multiple DEM differences over one glacier.

- 3.20: qualify here that SRTM is commonly used at regional-scales and over medium to long time periods as it is not exceptionally accurate and likely wouldn't be as much use for e.g. 2000-2001 mass balances.

  Added a clause, "though typically over longer time periods ($> 10$ year separation between DEMs)." to this sentence.

- 3,24-30: due to these problems, would it not have been better to select a region for which two high-quality regional-scale DEM products exist? Say, Iceland?

  Perhaps, but finding high-quality, regional scale DEM products with known dates is not an easy task. For example, the Iceland National DEM has significant errors/interpolation artefacts, as many areas are

interpolated from old topographic contours. While the glacier surfaces may be quite good, these artefacts and errors make estimating the uncertainty in the calculated volume changes much more difficult. In response to another reviewer, we have provided additional information about the size of the area impacted by these voids.

- 4,11-12: what's the justification for this omission now that we know that these very small glaciers are quite important? (Bahr & Radic, 2012, Cryosphere, doi:10.5194/tc-6-763-2012).

  We omit these smaller glaciers because errors/inaccuracies in glacier outlines are much larger for smaller glaciers. As our goal is to investigate the effects of void interpolation methods on estimated volume changes, it is best to have a larger sample of on-glacier pixels to work with; voids over small glaciers result in more limited data from which to extrapolate. Also, since our objectives are methods oriented, the question about small glaciers being important is not so relevant. For further comparison of results over small glaciers, we would suggest that higher spatial resolution DEMs are required as opposed to the medium resolution DEMs used here. We have attempted to clarify this in the manuscript. We now clarifiy this in the text.

- 5,1-2: specify most spaceborne stereo optical sensors'. Sensors onboard airborne platforms or historical aerial photographs will not have identical spectral range or resolution, and therefore may not be comparable with processing of ASTER scenes.

  Done.

- 5,13: mean and median, or the mean or median? Which? See also 6,7-8.

  Mean or median; changed to clarify.

- 5,20: if this is to be replicable then some more detail is required. Which surrounding pixels? Just those immediately proximal to the void? If so, this could be problematic as there may be inaccurate elevations just beyond the low correlation areas cutoffs. If not the very next pixel, then how many back from the void space? Provide enough detail of this method for another to reproduce your procedure exactly. See also 5,25

  The interpolation is carried out using `scipy.interpolate.griddata`, which triangulates the input data and performs linear barycentric interpolation (`https://docs.scipy.org/doc/scipy/reference/generated/scipy.interpolate.griddata.html`). We will make the scripts used to fill the voided DEMs, as well as a csv file of resulting volume changes, available through a github repository upon the acceptance of this paper.

- 6,26: why 10%? What's your justification?

  The 10% assumption is based on a conservative estimate of the error reported by the RGI (e.g., Pfeffer and others, 2014), found elsewhere in the literature (e.g., Brun and others (2017); Kääb and others (2012)). We have added these references to the text.

- 6,27: over what scales does spatial autocorrelation occur? I see this on the next page. But, why is it assumed to be 500 m (and why only 500 m given that it can occur on a range of scales simultaneously)?

  We have chosen 500 m based on the value used in other studies, including Brun and others (2017); Fischer and others (2015); Rolstad and others (2009); Magnússon and others (2016). We are aware that it could be smaller, but feel that 500 m is a good, conservative estimate based on this previous work.

- 18, Figure 2: Can you differentiate between the colour of the glacier outline and the ASTER correlation score mask? The middle panel all looks the same colour to me (except the red), even though I think its supposed to be dark grey outline and black mask.

  Done.

- 19, Figure 3: Shaded grey around elevation changes refers to uncertainties? If so, please state in the caption.

  Mean ± one standard deviation, now included in text.

- 7,9-10: why would you find the most voids occurring in the middle of the elevation range when from an optical image feature matching perspective (where the ASTER DEM gets its correlation score) you would expect fewer features and poorer correlation the higher up you go?

  For most of these glaciers, the higher elevations are on much steeper slopes with significantly higher contrast. The middle elevation ranges tend to be the flatter, more featureless parts of the accumulation area.

- 20, Figure 4: Background Landsat scene is a bit awkward to see as its so dark. Can you adjust the contrast, or similar to a previous comment, turn up the opacity to de-emphasise the background and emphasise the elevations changes? Looks like a graded colour scale, yet legend shows categories. Shouldn't the legend by a graded colour bar too? Likewise other figures.

  Regarding the color scale, see comments for Figure 1. We have changed the background to be a pan-sharpened Landsat scene with more contrast.

- 7,18: by acquisition area, do you mean accumulation area? If you're going to list individual glacier names in the main text, these need to be listed or shown in the figure somehow.

  In this case, we are referring to the 2012 and 2013 acquisition years for the IfSAR DEM, not the glacier accumulation areas. Glacier names are shown in Fig. 1, which we now refer to here.

- 7,24: I would say patterns' isn't quite the right word here. Some of the variability' perhaps?

  Changed.

- 7,25-26: is it therefore worthwhile to consider repeating this exercise at the local glacier (rather than regional) scale? And for simple vs complex perimeter glaciers?

  It could be interesting to consider the local glacier as well, but this will be heavily dependent upon each individuals glacier change with the amount and location of voids. In this study, the local scale is covered by about half of the extrapolation methods applied (See Figs. 5, 7, and 8). Furthermore, it is not necessarily the perimeter of the glaciers that's important here, it's the variability in elevation changes, where you have some surging/advancing glaciers, many heavily retreating glaciers that reach low elevations, and other glaciers that are also retreating, but don't necessarily have the same loss vs. elevation as larger glaciers due to dynamics.

- 23, Figure 7: Great figure, but for readability perhaps the RGI60.01.' part can be removed from each individual glacier on the y axis and be included in a single y axis label? Can you also indicate in the figure caption how the individual glaciers are sorted along the y axis? It doesn't appear to be by RGI ID number, or by volume change. Is it north-south, or by glacier area, or something else?

  Thank you. We have adopted your suggestions, and added "sorted by glacier area in descending order" to the figure caption.

- 24, Figure 8: It would be interesting to see this analysis extended to smaller glaciers, or the entire sample, but I understand if this is too time-consuming and therefore not possible.

  In general, the pattern is similar for the smaller glacier classes, just with more outliers.

- 9,1-20: some very small paragraphs here (comprising just one sentence sometimes). Is this necessary?

  We have combined the last two paragraphs, and added to the paragraph beginning at line 6.

- 9,18-20: can you add some value judgments between these best three, perhaps quantifying precisely how each do and therefore which performs best? Actually, nevermind that, I see it in the next paragraph.

  Never minded.

- 11,8: please quantify rather than just stating performed well'.

  Added "producing estimates within the uncertainty of the original estimates"

- 11,20-25: please replace do well', does well' etc, with perform(s) well'.

  Done.

**References**

Brun, F., E. Berthier, Patrick Wagnon, A. Kääb and Désirée Treichler, 2017. A spatially resolved estimate of High Mountain Asia glacier mass balances from 2000 to 2016, *Nature Geoscience*, **10**(9), 668–673.

Fischer, M., M. Huss and M. Hoelzle, 2015. Surface elevation and mass changes of all Swiss glaciers 1980-2010, *The Cryosphere*, **9**(2), 525–540.

Kääb, A., E. Berthier, C. Nuth, J. Gardelle and Y. Arnaud, 2012. Contrasting patterns of early twenty-first-century glacier mass change in the Himalayas, *Nature*, **488**(7412), 495–498.

Magnússon, E., J. Muñoz-Cobo Belart, F. Pálsson, H. Ágústsson and P. Crochet, 2016. Geodetic mass balance record with rigorous uncertainty estimates deduced from aerial photographs and lidar data  Case study from Drangajökull ice cap, NW Iceland, *The Cryosphere*, **10**(1), 159–177.

Pfeffer, W. T., A. A. Arendt, A. Bliss, T. Bolch, J. G. Cogley, A. S. Gardner, J. O. Hagen, R. Hock, G. Kaser, C. Kienholz, E. S. Miles, G. Moholdt, N. Mölg, F. Paul, V. Radić, P. Rastner, B. H. Raup, J. L. Rich, M. J. Sharp, L. M. Andreassen, S. Bajracharya, N. E. Barrand, M. J. Beedle, E. Berthier, R. Bhambri, I. Brown, D. O. Burgess, E. W. Burgess, F. Cawkwell, T. Chinn, L. Copland, N. J. Cullen, B. J. Davies, H. De Angelis, A. G. Fountain, H. Frey, B. A. Giffen, N. F. Glasser, S. D. Gurney, W. Hagg, D. K. Hall, U. K. Haritashya, G. Hartmann, S. J. Herreid, I. M. Howat, H. Jiskoot, T. E. Khromova, A. Klein, J. Kohler, M. König, D. Kriegel, S. Kutuzov, I. Lavrentiev, R. Le Bris, X. Li, W. F. Manley, C. Mayer, B. Menounos, A. Mercer, P. Mool, A. Negrete, G. Nosenko, C. Nuth, A. Osmonov, R. Pettersson, A. E. Racoviteanu, R. Ranzi, M. A. Sarikaya, C. Schneider, O. Sigursson, P. Sirguey, C. R. Stokes, R. Wheate, G. J. Wolken, L. Z. Wu and F. R. Wyatt, 2014. The randolph glacier inventory: A globally complete inventory of glaciers, *Journal of Glaciology*, **60**(221), 537–552.

Rolstad, C., T. Haug and B. Denby, 2009. Spatially integrated geodetic glacier mass balance and its uncertainty based on geostatistical analysis: Application to the western Svartisen ice cap, Norway, *Journal of Glaciology*, **55**(192), 666–680.

---

## Author Comment (AC2) · 25 Jan 2019

**Reponse to Reviewer 2**

McNabb et al. compare different strategies of filling data gaps or interpolating sparse measurements of glacier elevation change in order to obtain the best estimate of total volume change (and ultimately glacier-wide mass balance). They assess the relative performance of the different gap-filling methods by comparing their results to the true volume change from the complete map of elevation change, an assessment both at the scale of individual glaciers and at the regional scale.

This is a certainly welcome study and I foresee that it is going to be widely cited. Indeed, almost all studies performing geodetic mass balance estimates need to handle data gaps. The procedure to assess the influence of different gap-filling method (i.e. taking a complete map of elevation change dh and creating realistic data voids in it) is adequate. That said, I was somewhat disappointed by the paper. It is not always clear and the writing could be improved. More importantly, I ended up with some questions that, I think, could have been, at least partly, answered. More work is needed to fully exploit this nice dataset and to transform this good study in a benchmark paper for the community.

We would like to thank the referee for their careful and constructive comments that have helped to improve both the clarity and focus of the manuscript, and we hope our revisions will be satisfactory. Our responses to the comments below are in blue, with the original comments in black.

**1   General comments**

1. Choice of unit to report the results. The authors have chosen to report their total volume change (and their departure from the true value) in km 3 . I do not find this unit really useful, as it is so much dependent on the glacier area. This is why, most studies use the very convenient unit of m w.e. yr -1 (or kg m -2 yr -1 ) to report mass balances. With the latter unit, it is easy to compare different glaciers within a region or glacier mass balance from different regions. I fully understand (and support the fact) that the authors do not want to provide mass balances here because many additional corrections would be required to obtain a meaningful value. Thus, I suggest that they use glacier-wide or region-wide elevation change (thus in meters), together with % of error (as already done).

   In accordance with this comment, as well as the comments from another reviewer, we have re-presented the analysis using $\mathrm{m\,a^{-1}}$, that is, the area-averaged rate of volume change.

2. How to handle data gaps in the error estimate. A missing section/discussion is how to take into account the data gaps in the formal error estimate. Right now, authors performed a sound sensitivity analysis and conclude on the best strategies, which is already useful. However, a remaining question is how to include the uncertainties dues to data gaps in the formal error estimate. I do not think this is done well in the literature so far and I was hoping to find an

answer here. Authors would increase the impact of their work if they could provide, at least, some suggestions. I know this is not straightforward but really hope they can tackle this issue.

We agree, both that this is not done particularly well in the literature, and that this would be a very good addition to the literature. Unfortunately, we are unable to include a correct, formal handling of the uncertainty introduced by data gaps, but rather provide insight into the magnitude of the potential error given a certain void percentage. We hope that the improved section 4.5 and Figs. 10-12 in the updated manuscript provide some useful insight on this topic. The formal handling of this error must somehow include also the ability to guess at the missing changes, and will depend strongly upon each individual dataset. For example, a dataset with a few random measurements over a geometrically simple accumulation area may be less uncertain than over a more complex accumulation area with multiple basins where both accumulation and dynamics will be spatially varying. Therefore, also at this point in time, we are not sure how to best formulate an extrapolation error equation for data gaps.

3. % of data gaps. The gap creating method makes sense. However, I had the feeling that the % of data gaps was not very high and the data voids not large. Are these percentages of data gaps in line with published values? A more aggressive gap creating threshold is discussed, but too briefly. How much data gaps are created in this case? I think many readers would be curious to know if the conclusions hold when 50% (or more) of data gaps are present.

In the updated manuscript, section 4.5, we have gone into much greater detail, using multiple correlation thresholds to analyze the differences for individual glaciers. In particular, we examine the performance vs. percent void for the best-performing methods chosen based on the 50% threshold case.

4. Variability of dh in the study region. I miss a more thorough description of this variability. This is important here because in an end-member case (hypothetic) where there would be no spatial variability of elevation change, then most gap-filling methods would work well. How does variability vary with elevation? I expect less dh variability at high elevations where data gaps tend to be concentrated, which may explain why the local hypsometric approach works well. To quantify variability, individual glacier mean elevation change (not glacier-wide mass balance) could be calculated and the spread shown. How does this spread compare to earlier studies? It would also help to discuss whether the study region is representative.

This is a welcome suggestion, and we have included this discussion in its own section, 4.2, which looks at both the variability within the region and compared other published values. This part of Alaska seems to be somewhere in the middle of values for regions including other parts of Alaska, the Arctic, and High Mountain Asia. Fig. 3 in the text shows the variability within (normalized) elevation bands, which confirms your expectations here - less variability at higher elevations, higher variability at lower elevations.

5. Global hypsometric approach, normalized elevations or not? To take into account the diversity of the altitude range of glaciers in a region, some earlier studies have normalized the elevation in order to extrapolate to un-surveyed areas. This is also what the authors do here to plot dh in their Figure 3. I was wondering if the normalization helped or not for the extrapolation. This procedure seems to make sense and it would be good to test its added value.

Other studies, such as Arendt and others (2006), have looked into this with more detail. We discuss this point in more detail at 9.31-10.2, indicating estimates may be improved for individual glaciers

by using the normalized elevations; however, we obtained good results using absolute elevation changes with the global mean hypsometric method, which may indicate that it is not needed.

**2 Specific comments**

- 1.1 mass balance does not imply sea level. Glacier mass gain/loss does.

  Changed.

- 1.2 Mentioning glaciological measurements in the abstract is not really useful. Not the core of the paper.

  Removed this sentence, replaced with "Recently, glacier mass balance has been estimated on individual glacier and regional scales using repeat, full-coverage digital elevation models (DEMs)."

- 1.5. Is "based" the best word here?

  Changed to read "the properties of which depend on..."

- 1.18. One further and strong important limitation of the glaciological mass balances is that they seem to be performed on glaciers where the mass balances tend to be more negative than the regional average (Gardner et al., Science, 2013).

  Indeed, and we have added a sentence here to reflect this.

- 1.20 They must be a reference for the WGMS data and also for the number of glaciers on Earth

  RGI reference added for the number, and WGMS reference added.

- 1.22 A reference to a review? Possibilities I see are:

  - Bamber, J. L. and Rivera, A.: A review of remote sensing methods for glacier mass balance determination, Global and Planetary Change, 59(14), 138148, doi:10.1016/j.gloplacha.2006.11.031, 2007.
  - Bamber, J. L., Westaway, R. M., Marzeion, B. and Wouters, B.: The land ice contribution to sea level during the satellite era, Environmental Research Letters, 13(6), 063008, 2018.
  - Marzeion, B., Champollion, N., Haeberli, W., Langley, K., Leclercq, P. and Paul, F.: Observation-Based Estimates of Global Glacier Mass Change and Its Contribution to Sea-Level Change, Surveys in Geophysics, 38(1), 105130, doi:10.1007/s10712-016-9394-y, 2017.

  We have included a reference to Bamber and Rivera (2007) here.

- 1.25. Do the authors exclude from the geodetic method (and thus from the study) all ICESat-based estimates of glacier volume change? ICEsat provides sparse measurements that need extrapolation. To be clarified.

  Added a parenthetical statement to make clear that we aren't excluding laser alimtetry/ICESat studies from the 'geodetic' label.

- 2.6. Acronym DEMs to be used here, as defined already. Do the authors understate that they exclude estimate based on ICESat or sparse GPS surveys?

  It was not our intent to exclude ICESat surveys from this consideration, and we have updated the text accordingly.

- 2.9 Maybe a short statement that this is certainly true for old imagery (8-bits) but that this issue is strongly reduced using state-of-the-art 11- or 12-bits stereo data? In the end, I also note that the data gaps are not so concentrated in the accumulation area.

  added "though this problem has been reduced with improved radiometric resolution of more modern sensors." to the end of this sentence.

- 2.25 I think the interpolation methods should be described only once but not "briefly". They are the heart of the study.

  The methods are described in more detail in the actual body of the paper, but we feel that a brief summary is appropriate here.

- 2.28. Does it make a difference that the elevation with altitude is used to fill unsurveyed values vs. just multiplied by the area of the altitude band? For the glacier-wide mass balance (or the glacier-wide dh) I think it is the same. Maybe state it to avoid confusion for some readers.

  Changed to read "and estimating elevation change as a function of elevation, integrating this curve with the glacier hypsometry", as the practical difference between the two approaches is at most very small.

- 2.30 I very strongly suggest using regional instead of global. I found global confusing (I immediately thought about the whole Earth). Or did I miss a difficulty linked to the use of "regional"?

  We have chosen to use "global" and "local" terminology based on the same terms used in "global" and "local" methods of interpolation. In the former, a single function is used to interpolate over the domain of a dataset, whereas in the latter, the function chosen changes based on the properties of a subset of the data. We have now stated this in the text.

- 2.31 "basin" needed after "glacier"?

  Removed, and elsewhere.

- 2.35 The sentence "In this paper, we use two high-quality, radar-derived DEMs." does not appear to be complete and break the flow of the introduction.

  We have moved this sentence to the end of the paragraph, and added more information about how we perform the study.

- 3.14. I think the key point for this study is that the authors have a large intra-glacier and inter-glacier variability of elevation change (a consequence of the variety of glacier type). Make it clear and quantify better (see general comments). The authors may note that some previous workers have separated different glacier types while extrapolating.

  Sentence changed to read "As such, it is an ideal region to estimate the effects of using spatially incomplete DEMs to estimate glacier volume changes, as it provides a diverse sample of glacier types, sizes, and altitude ranges, with a high variability of intra- and inter-glacier elevation changes."

- 3.24. % of data gaps in SRTM for this study area?

  Fewer than 2.5% for the glacierized area; we have added this information here.

- 4.26 Could also have been done on the SRTM. Maybe state that this is an arbitrary choice.

  True, and stated.

- 5.3 How did the authors handle clouds in ASTER?

  ASTER scenes were chosen based on being mostly cloud-free over glaciers. If clouds were present in the images over glaciers, this is reflected in low correlation scores.

- 5.9 As said before, description of each interpolation method is central to the study. So we do not want to have a "brief summary" only. In fact the description is detailed enough.

  Removed "brief" from this sentence.

- 5.14 Here and elsewhere I found the use of "glacier basin" instead of "glacier" a bit problematic. For me a glacier basin includes the glacier + the off-glacier terrain included in this basin. Why not using "glacier" simply ? (everywhere)

  To reduce confusion, we have removed references to 'glacier basins' throughout the paper, and replaced with either 'glacier outline' or simply 'glacier'.

- 5.22 "linear interpolation". Should not it be "bilinear"?

  Yes, it should be.

- 5.22 "because the voids are relatively small" is not a very precise statement. It lack quantification (void size?) and one also would like this study to address the case of large data voids.

  We have removed this statement.

- 6.8 is 'original elevation' clear enough?

  Changed to "elevation in the earliest DEM"

- 6.27. IMPORTANT. I see no reason why the systematic error in elevation difference (epsilon_bias) obtained using triangulation between the DEMs should be divided by the square root of the number of effectively independent pixels. Either justify or correct.

  Thank you for catching this typ-o. We have fixed this equation and the following one.

- 7.8 I would have expected a higher percentage of voids in the accumulation area. This is not the case. This should be discussed.

  It is actually the case, though. The bulk of the area-altitude distribution corresponds to the relatively flat, mostly featureless portions of the accumulation areas, while higher elevations tend to be on much steeper slopes where there is more contrast in the ASTER scenes (which leads to higher correlations).

- 7.14 title of section 4.2 is not really meaningful. Improve section and sub-section titles if possible.

  Changed to "Impacts on Individual Glacier Estimates"; changed 4.3 to "Impacts on Regional Total"

- 7.16. An elevation change can be negative, not a pattern.

  Changed to read "elevation changes in the region are negative, especially at lower elevations."

- 7.21 " The pattern of elevation change shown on Rendu Glacier in the elevation difference maps". Authors need to improve the text.

  replaced "the elevation difference maps" with "Fig. 4"

- 7.26 to 8.2. These sentences are not really well written and the reasoning is hard to follow. In fact, I do not see the rational for using volume change in km$^3$ (and quoting an average volume change). This unit is so much dependent on the size of the glaciers whereas the global hypso method consist (if I understood correctly) in using mean/median dh per elevation band. So if a glacier (whatever his size) as a dh vs. altitude pattern like the rest of the region then the method should work.

  We have updated the text to use units of m a$^{-1}$, and indicated the variability in glacier elevation changes as suggested previously.

- 8.4 the fact that the authors do the conversion here to average elevation change (in meter), nicely illustrates the limit of the total volume approach (in km3).

  We have removed this paragraph.

- 8.7. IMPORTANT. The fact that the authors interpolate "over much smaller areas" (and the authors are aware of that) is quite problematic. It suggests that the authors are in a configuration (with sparse data voids) where local gap filling methods will all perform reasonably well. A much more aggressive gap creating strategy should be considered in an alternative scenario.

  Kääb (2008) used contour lines, and differences at contour lines, to interpolate a DEM, hence our use of "much smaller areas". We have removed this paragraph in the updated text, and greatly expanded the discussion and analysis in section 4.5 to show much more aggressive cases.

- 8.8-10. I do not follow the reasoning. Contour lines are maybe (certainly) biased at high elevation but a DEM created from them does not have data gaps. So the fact that contour line is floating is a different problem (like radar penetration) and does not influence the errors due to gap filling. Or better explain if I missed something.

  As above, we have removed this paragraph.

- 8.30 Showing the dh with altitude for each of these 20 glaciers and the regional mean value would nicely illustrate the text.

  Thank you for the suggestion. With the number of figures we have, we feel that it would not be an effective use of space to show this here as well.

- 9.11 Did authors used the term "global fits" before. I do not think so. If they want the readers to follow them, then they need to stick to a terminology.

  Changed to use "methods", in line with the previous terminology.

- 9.22 "Differences" of what?

  Differences to true values. Text updated.

- 9.26 authors need to clarify "relative". Is it normalized? If yes, I think they should quantify the added value of the normalization for the same global mean hypsometri method.

  Updated text to use 'normalized' in place of 'relative'.

- 9.30 "one explanation for the value". Do the authors want to discuss a high/low value? Clarify. Did they expect this method to perform better? Avoid such understatements.

  Changed to read "the overall worse performance of the elevation interpolation method versus linear interpolation of elevation change", as this is the comparison we intended to make.

- 10.9 do the authors suggest using the median rather than the mean as a metric of centrality for an elevation bin? I think it could be dangerous because the dh distribution could also be quite skewed with an elevation bin (when it comes to large glaciers for example). At least this needs to be discussed.

  We suggest that using the median for an elevation is less bad than using the median for an entire glacier; that said, it clearly does not do as well based on our results. We have added a sentence at the end of this paragraph to clarify this point.

- 10.11 Authors need to provide the corresponding % of data gaps? Does this more aggressive threshold really lead to a strong increase in data gaps? Where on the glacier?

  Please see the updated text, which discusses these issues in much greater detail.

- 10.22 Authors should detail how the ASTER DEMs were derived. Depending on the methods (and correlation threshold) the percentage of data gaps will change quite a lot. The following question is thus raised: Is it better to keep only the most reliable values in the DEM and increase data gaps (and filled them afterward) or alternatively try to get the DEM processing parameters resulting in the most complete DEM. If the authors could also contribute to this research question they would increase the impact of their study.

  We have added more details to this paragraph, and attempted to answer the bigger question at the end of this section.

- 10.25 was dDEM defined already? (not sure)

  Yes, at 5.24.

- 10.29 Is this value of 0 km$^3$ the volume change estimate, suggesting surprisingly no volume change? Or the difference to the true IfSAR/SRTM value?

  This is the volume change estimate, not the comparison to the "true" value. We have added 'total' to this sentence to help clarify this point.

- 10.31 "3.6" positive value of volume change? OK?

  That's the value that we get, yes.

- 11.1 surprising statement that the two methods perform as well when authors just illustrated the danger of the linear interpolation method...

  Added 'in the idealized case presented here' to clarify.

- 11.25 the dependence on the size of the voids has unfortunately not been examined sufficiently.

  We hope that the updated section 4.5 is sufficient.

- 11.30 I do not think this issue of proximity has been really addressed so that such a conclusion can be made. Or I misunderstood the statement? Do the authors suggest using a modified global method using only the glaciers in the vicinity of the one for which volume change needs to be calculated?

  This statement covers the local hypsometric methods, as well as the spatial interpolation methods, which use values either from a given glacier, or from areas within a small area around the glacier in the case of glacier complexes. We have clarified this in the text.

- 11.33 "suffice" well anyway there is no other choice right? If only a few "anomalous" glaciers are sampled than the regional total could be strongly biased

  Added a sentence 'Additionally, the regional bias for such a case may be strongly biased, as discussed in previous studies.' to further clarify this point.

- Table 1. Can the authors tell if these are simple (as I guess) or area-weighted statistics? Maybe remind in the legend the number of individual glaciers on which these statistics are obtained.

  These are not area-weighted statistics, but 'simple' statistics as you have guessed. We have included the total number of glaciers in the legend, as suggested.

- Figure 1. I could not find name of glaciers on this figure.

  We have increased the font size to make this easier to read.

- Figure 3. What is the envelop around the mean/median dh? 1-sigma of data?

  Yes. This is now indicated in the figure caption.

- Figure 2. An extra panel showing the distribution of data gaps for the more aggressive correlation threshold would be welcome. Also provide on each panel the % of data gaps for Taku Glacier.

  We have provide the percentage of data gaps for Taku, as suggested, but have not added a panel to the figure.

- Figure 5. The authors use Actual volume change here but true volume change in the text. Homogenize. Are all the acronyms used to name the different methods in the figure defined (in the text or the legend)?

  Label change to use "true" instead of "actual." Acronyms are now introduced in section 3.2, and kept consistent throughout the figures and tables.

- Figure 6. Rather than showing the dh maps for all methods (with some maps that are very similar), it would probably be best to show only the ones with strong difference. Also it would be good to show the map with data voids. So that the reader as a good sense of where the voids where. Authors could also consider moving this figure (or the suggested revised version of it) to the supplement. Showing instead the pattern of change with altitude for Taku derived from these maps could likely better illustrate some of the subtle differences mentioned in the text.

  By including all maps, even those that look very similar, we illustrate that in some cases, there aren't many differences between the methods investigated. We have included the data voids in panel a, but have otherwise left the figure unchanged.

**References**

Arendt, A. A., K. A. Echelmeyer, W. D. Harrison, C. S. Lingle, S. L. Zirnheld, V. B. Valentine, J. B. Ritchie and M. Druckenmiller, 2006. Updated estimates of glacier volume changes in the western Chugach Mountains, Alaska, and a comparison of regional extrapolation methods, *Journal of Geophysical Research*, **111**(F3).

Bamber, J. L. and A. Rivera, 2007. A review of remote sensing methods for glacier mass balance determination, *Global and Planetary Change*, **59**(1–4), 138148.

Kääb, A., 2008. Glacier Volume Changes Using ASTER Satellite Stereo and ICESat GLAS Laser Altimetry. A Test Study on Edgeøya, Eastern Svalbard, *IEEE Transactions on Geoscience and Remote Sensing*, **46**(10), 2823–2830.

---

## Author Comment (AC3)

**Reponse to Reviewer 3**

In this paper, McNabb et al. investigate the effect of missing data (called voids in the article) on the glacier volume change that can be obtained from digital elevation models (DEMs) differencing. The methodology is rather straightforward, they differentiate two DEMs acquired over Southeast Alaska that (almost) cover the entire glacierized area. These data are used as reference data. Then, they artificially generate voids in the data and evaluate the impact of different void-filling/interpolation methods on the on the regional glacier volume change estimate, but also for each individual glacier. They investigate 11 different void-filling/interpolation methods that are often used in the literature, providing a unique and comprehensive assessment. They conclude that most interpolation methods introduce very little bias (<1%) on the regional glacier volume change. However, individual glacier volume change estimates can be severely affected by the choice of the interpolation strategy.

This paper is rather narrow-focused, but its scope fits very well within The Cryosphere, where it will certainly reach an adequate audience. The topic is timely and very relevant, as the geodetic method is more and more widespread in glaciology. To my opinion, this paper has the potential to become a classic paper in the field of geodetic mass balance. However, and while I appreciate the concision of the paper, I have the feeling that the authors could discuss some aspects more in depth. Moreover, I sometimes had a hard time following the paper and found that it lacks clarity in its current form. These are my two major comments.

We would like to thank the referee for their careful and constructive comments that have helped to improve both the clarity and focus of the manuscript, and we hope our revisions will be satisfactory. Our responses to the comments below are in blue, with the original comments in black.

**1 Major comments**

1. **Volume change vs. geodetic mass balance**
   The title of the paper mention the sensitivity of geodetic glacier mass balance, but actually discuss only glacier volume changes. This decision is somehow understandable, because it is the quantity that is directly affected by the void-filling strategy. However, the impact of void-filling strategies on the individual glacier volume change expressed in $km^3$ is not very intuitive, and not as informative as it could be. First of all, the IfSAR DEM was acquired over two years and it would be better to present the annual mean instead of the totals, in order to get rid of this temporal inconsistency. Second, the results are largely dependent on the glacier area considered, larger glaciers being more sensitive to the interpolation (P8L12-13), mostly because for a similar elevation change they have larger volume change, due to their larger area. For example, for figures 7 and 8 (and 5?), I suggest to present the results in $kg\ m^{-2}\ a^{-1}$ or in $m\ a^{-1}$ (if the authors do not want to make any density assumption).

   In accordance with this comment, as well as the comments from another reviewer, we have rewritten the analysis using units of $m\ a^{-1}$, that is, the area-averaged rate of volume change.
   If the authors want their study to be reproduced and the conclusions of this article to be applied elsewhere, they need to analyze the influence of the void-filling strategy more in depth. I feel like the paper misses some basic, yet interesting analysis. For instance, what is the influence of the percentage of voids for individual glaciers? Of the glacier-wide mass balance/mean rate of elevation change? Of the glacier area?

The authors probably analyzed these influences already and found that they were limited/not interesting, but I think it is probably worth mentioning them, in order to apply their conclusions to a different setting.

We have significantly expanded the discussion on the impact of voids for individual glaciers and methods in section 4.5, in particular adding Figures 10, 11, and 12, and focusing on the percentage of voids where the 'best' methods start to give unreliable results. We hope that this helps to increase the impact and depth of the analysis.

2. **Some clarifications needed**

The objective of the study is quite straightforward, but a number of confusions and unclear statements prevent from an easy understanding of the paper. I had to go back and forth a number of time reading the paper, and I have the feeling that the clarity of the paper could be much improved if the authors address the three comments below. First, the author mimic the voids of a standard DEM difference, based on ASTER correlation map patterns. Consequently, I expected that they would investigate the influence of the void-filling strategy for this purpose. However, they also investigate such methods as the global ones, which are generally used for regionalization of Lidar surveys. They should make a clear distinction between these two applications when relevant. In other words, I do not think that the global methods are relevant for DEM differences void filling at the scale of individual glaciers. Correct me if I'm wrong, but I do not know any paper which studied individual glacier mass balances obtained with such global methods to fill in the holes of a DEM difference.

You are correct in saying that the global methods have not really been used to estimate individual glacier mass balances in prior studies. Our goal with including these methods was to investigate whether useful results could be obtained for an individual glacier using global methods, as well as to compare the regional estimates obtained using a variety of methods. In the updated text, we have tried to make this distinction more clear, in particular at P6L20-24.

Second, the different methods described are relatively basic, however their description should be clearer. For instance, adding equations to the description of each method would be beneficial. Alternatively, you could share the code you wrote, which would also support your conclusion in which you encourage others to test different methods when dealing with voided data.

We have added a statement in both the Methods section, and the Code Availability about where the scripts used to generate and interpolate the voids will be available.

Third, I found the example about Taku Glacier extremely confusing. If I understand correctly figure 2, the tongue of Taku Glacier is mostly free of voids. However, the global methods (panels e to g) totally change the pattern of areas where data are available! Consequently, the methods should be described as Interpolation and not Void-filling (for instance the title of section 3.2 should be changed), because they also apply to areas without voids (and it is technically a non-exact interpolation method). If I did not understand correctly figure 2, you can ignore this comment, but you should consider changing figure 2.

You are correct that the tongue of Taku Glacier is mostly free of voids; you are also correct that these methods would be better described as 'void interpolation' rather than 'void filling', given that we aren't actually 'filling' the voids, per se. We have updated the text to reflect this.

**2  Specific comments**

- Fig. 2: confusion between the glacier and voids outlines. You should draw the glacier outlines in a different color/line thickness, such as the panel 3 is easier to understand.

  We have changed the color of the voids in Fig. 2 in order to avoid this confusion.

- Fig. 5: add a scale/grid on the inset.

  The updated figure no longer has an inset.

- Fig. 6: this figure is extremely confusing to me. First of all, I'm missing the location of the voids on panel a, and I had to go back on the previous figures to understand where the voids where. Then the figure shows the strong impact of the void-filling procedure on the tongue of Taku Glacier, but from what I understood of figure 2, there were no voids on the tongue. This comment is in line with my major comment 2.

  See response to your major comment 2 above. We have shown the void locations in the updated Figure, panel a, to help make this figure more clear.

- Fig. 7: I think here the reader loses the information about the difference to truth relative to the total glacier volume change. I guess the larger the volume, the larger the error.

  The updated figure shows the differences averaged by glacier area, rather than the volume change in $km^3$, in accordance with other comments.

- P1L3 and P1L17: I do not fully agree with your definition of the glaciological method, which does not really monitor changes in surface height (that would actually be the geodetic method). The glaciological method directly measures the surface accumulation and melt.

  Changed to read, "Traditional estimates of glacier mass balance have involved *in-situ* seasonal or annual measurement of accumulation and ablation at select locations, and extrapolation of these sparse measurements to the entire glacier..."

- P1L8-9: add a word about the artificially generated voids

  Done.

- P1L11: define ASTER

  Done.

- P2L6: Digital Elevation Models-¿ DEMs

  Done.

- P2L33-P3L6: this paragraph is not really well structured in my opinion. You should describe more clearly the philosophy of your study. I suggest to move completely your warning statement about the radar DEM difference (P3L1-2) to the other place where it is mentioned (P4L19-24). You need to add something about the artificially generated voids and to better justify the choice of ASTER, among a large choice of (optical) sensors.

  We have moved the warning statement to the end of the paragraph, and added something about artificially generating the voids used. We feel that the warning is important to have in the introduction, as well as later on in the paper. We don't feel that the choice of ASTER needs justification here, as it is a widely-used sensor in glacier studies with a long (nearly 20-year) record.

- P3L9-14: what is the glacierized area?

  ~$5900\,km$ $^2$, added to text.

- P3L19-20: provide references about studies that used SRTM to estimate geodetic glacier mass balance

  Done.

- P3L29: what proportion of the glacierized are if affected by these small voids?

  Fewer than 2.5% of the glacierized area; we have added this information here.

- P4L2-6: more references and details are needed in this paragraph. What is the precision of the IfSAR DEM? What is the proportion of voids? Has it been used in other glaciological studies?

  We have added more information here. The metadata report an accuracy of ∼1 m, while the USGS-reported accuracy is 3 m. Voids in the original acquisitions are small (<1% according to the metadata), and are filled in post-processing using proprietary algorithms.

- P4L5-6: is this sentence useful?

  Probably not, so we removed it.

- P4L12: can you justify the exclusion of glaciers smaller than 1 km$^2$ from your analysis?

  We omit these smaller glaciers because errors/inaccuracies in glacier outlines are much larger for smaller glaciers. As our goal is to investigate the effects of void interpolation methods on estimated volume changes, it is best to have a larger sample of on-glacier pixels to work with; voids over small glaciers result in more limited data from which to extrapolate. Also, since our objectives are methods oriented, the question about small glaciers being important is not so relevant. For further comparison of results over small glaciers, we would suggest that higher spatial resolution DEMs are required as opposed to the medium resolution DEMs used here. We now clarifiy this in the text.

- Method section: at some point you need to explain how you calculate the regional estimate. Is it the sum of the individual glaciers, or do you consider all the glaciers as a single body of ice? Do you include the glaciers smaller than 1 km$^2$ in this estimate? If not you might bias it.

  It is the sum of the individual glaciers. We have included this to help clarify this point.

- P5L9: see my major comment 2 -¿ you might want to rename this paragraph Interpolation instead of Void Filling

  Changed to "Void Interpolation"

- Void-filling section: how do you deal with the temporal inconsistency between your two IfSAR DEMs? I guess for most method you interpolate the rate of elevation change and not the elevation change? This should be written clearly. However, this is not possible for the method based on the interpolation of elevation.

  As stated in section 2.2.3, we remove any glacier outlines that fall 10% by area or more in both collection years, to avoid this problem, adjusting the area of the remaining glacier outlines that fall within both DEM acquisition years accordingly.

- P6L4-6: here the reader wonders why using ASTER voids for regionalization applications (i.e., Lidar based studies)? In order to test the influence of regionalization, one could extract elevation changes in your DEM difference along Lidar flight lines... but this would be another paper!

  Indeed, and it has been done by other papers, for example Berthier and others (2010). Here, we are attempting to show that while these methods may not be a useful way of estimating the mass balance of an individual glacier, it can still be useful to estimate the mass balance of a region using data from the glaciers where you do have measurements.

- P7L6: define normalized glacier elevation

  Added "(i.e., the elevation divided by the elevation range)"

- P7L14: consider switching the order between the sections 4.2 and 4.3.

  We prefer the section order as-is.

- P7L15-23: this paragraph is rather disconnected from the rest of the analysis.

  We disagree. This paragraph discusses the variability of elevation changes in the region, which helps to explain some of the differences seen between the methods.

- P7L16: the pattern of elevation change is negative -¿ the phrasing is not clear to me

  Changed to read "elevation changes in the region are negative, especially at lower elevations."

- P7L26-27: the average geometric volume change has probably little influence, contrary to the mean elevation difference.

  Per your and another reviewer's suggestions, we have changed the units from $km^3$ to $m\,a^{-1}$.

- P8L3-10: can you say a word about the constant methods? And about the 1km neighborhood method?

  Added in the updated text in the following paragraphs at lines .

- P8L4: define RMS. This sentence is not completely clear to me.

  We have removed this paragraph. RMS values are now defined in the text at P8L16.

- P8L13-14: the larger glaciers are more sensitive than the others, because they have larger volume change for a similar elevation change, due to their larger area. This is one on the limitations of your analysis, because you look at volume changes only (see my major comment 1). You should consider extending your analysis to glacier mass balance, or rate of elevation change.

  Please see the updated text, which compares the area-averaged volume changes as suggested.

- P9L6-8: this paragraph is very short, while Fig. 8 is probably a key figure! Could you elaborate a bit? I found an unbalance with the previous paragraph (P8L30-P9L5), which is less important and much longer than this one.

  We have expanded the discussion of this figure, including adding analysis for smaller glaciers as well.

- P9L20: the 1km neighborhood is never mentioned earlier in the section 4.2.

  We now mention it in the paragraph discussing the updated Fig. 8.

- Section 4.3: consider adding a column to Table 1, which summarizes the regional totals for each method. It might also be interesting to discuss the difference between the regional estimates obtained by summing the individual glaciers versus the regional totals obtained by applying each method to the glacierized area considered as a single body of ice.

  Given that the goal is to show how the regional total changes using each method, we feel that the column showing the change is sufficient here. I'm not sure I understand the second point here - the regional estimates are achieved by summing the volume change estimates for each glacier, not by applying the methods as though the total area were a single body of ice.

- Section 4.4: why don't you use the percentage of voids of individual glaciers to study the influence of the percentage of voids on the distance to truth?

  See updated text (now section 4.5), as well as the newly-added figures which discuss this in much greater detail.

- P10L12: give the total percentage of voids for each threshold.

  See the newly-added Figure 10, which shows this.

- P10L28: again I got confused because you look at 91 individual glaciers and then you mention the global mean hypsometric as one of the best performing method... Please clarify.

  The global method uses elevation differences from all of the glaciers in a sampled area and is applied to each glacier based on that glacier's hypsometry. Hence, the global method here is only using the data from these 91 glaciers.

- P10L31: you can mention in the text that outliers are more often located near the voids, which increases their influence in a linear interpolation.

  A good point, which we have included in the text.

- P11L5-7: the order (regional volume change then individual glacier volume) is the opposite from section 4.2 and 4.3.

  Re-written to change the order.

- P11L25-26: you actually did not demonstrate this in your analysis...

  Hopefully, you agree with us that the updated analysis serves to demonstrate this.

**References**

Berthier, E., E. Schiefer, G. K. C. Clarke, B. Menounos and F. Rémy, 2010. Contribution of Alaskan glaciers to sea-level rise derived from satellite imagery, *Nature Geoscience*, **3**(2), 92–95.